# SOX17 regulates uterine epithelial–stralk acting via a distal enhancer upstream of *Ihh*

Xiaoqiu Wang [1,7], Xilong Li[2,8], Tianyuan Wang[3], San-Pin Wu[1], Jae-Wook Jeong[4], Tae Hoon Kim[4], Steven L. Young[5], Bruce A. Lessey[6], Rainer B. Lanz[2], John P. Lydon[2] & Francesco J. DeMayo[1]

Mammalian pregnancy depends on the ability of the uterus to support embryo implantation. Previous studies reveal the *Sox17* gene as a downstream target of the *Pgr*-Gata2-dependent transcription network that directs genomic actions in the uterine endometrium receptive for embryo implantation. Here, we report that ablating *Sox17* in the uterine epithelium impairs leukemia inhibitory factor (LIF) and Indian hedgehog homolog (IHH) signaling, leading to failure of embryo implantation. In vivo deletion of the SOX17-binding region 19 kb upstream of the *Ihh* locus by CRISPR-Cas technology reduces *Ihh* expression specifically in the uterus and alters proper endometrial epithelial–stromal interactions, thereby impairing pregnancy. This SOX17-binding interval is also bound by GATA2, FOXA2, and PGR. This cluster of transcription factor binding is common in 737 uterine genes and may represent a key regulatory element essential for uterine epithelial gene expression.

[1] Reproductive and Development Biology Laboratory, National Institute of Environmental Health Sciences, Research Triangle Park, NC, USA. [2] Department of Molecular and Cellular Biology, Baylor College of Medicine, Houston, TX, USA. [3] Integrative Bioinformatics Support Group, National Institute of Environmental Health Sciences, Research Triangle Park, NC, USA. [4] Department of Obstetrics and Gynecology and Reproductive Biology, Michigan State University, Grand Rapids, MI, USA. [5] Department of Obstetrics and Gynecology, University of North Carolina, Chapel Hill, NC, USA. [6] Deptartment of Obstetrics and Gynecology, University of South Carolina School of Medicine, Greenville, SC, USA. [7] Present address: Department of Animal Science, North Carolina State University, Raleigh, NC, USA. [8] Present address: Feed Research Institute, Chinese Academy of Agricultural Sciences, Beijing, China. These authors contributed equally: Xiaoqiu Wang, Xilong Li.  Correspondence and requests for materials should be addressed to F.J.D. (email: francesco.demayo@nih.gov)

The steroid hormone progesterone (P4) acts via its cognate receptor PGR to regulate uterine receptivity for embryo implantation. PGR regulates uterine epithelial cell proliferation, the epithelial acceptance of embryo invasion, and decidual differentiation of stromal cells in support of embryo development[1]. Indian hedgehog (encoded by *Ihh*) expressed in the uterine epithelium is under the control of P4 via PGR as well as its signaling modifier GATA2, contributing to uterine compartmental cross-talk required for epithelial–stromal interaction and facilitation of a successful pregnancy[2–5]. Identification of factors modulating the activity of the PGR is critical in identifying the molecular mechanisms regulating fertility. Recent studies using a genome-wide transcriptomic approach have identified SOX17 as a regulator of PGR action in the uterine epithelium[5].

SOX17 (encoded by *Sox17*) is a transcription factor belonging to the family of sex-determining region Y-related high-mobility group box, namely SOX. The mouse and human genomes each encode for 20 *Sox* genes[6], many of which (*Sox2, Sox9, Sox17, Sox4,* and *Sox18*) are key determinants of cell identities and regulate the capacity to reprogram the cell fates by pioneering the epigenetic remodeling of the genome[7–13]. To fulfill such a role, SOX transcription factors can specifically bind and bend DNA with angles ranging from 60° to 70°[14–16]. In particular, SOX17 regulates embryogenesis[17–20], maintenance of fetal and neonatal hematopoietic stem cells[21], segregation of ventral foregut endodermal organs[22–24], and cardiovascular development[25]. Recently, SOX17 has been identified as the target of PGR[5] and the novel mutated gene in uterine corpus endometrial carcinoma[26], suggesting the potential role of mediating PGR action in the uterus during pregnancy and disease states.

Female mice with loss of one *Sox17* allele demonstrated reduced fertility[27]. It has been reported that mice with ablation of *Sox17* in PGR-positive cells do not have uterine glandular structures and are infertile[28]. Similarly, ablating *Sox17* in uterine epithelia also results in infertility[28]. Here, we provide mechanistic insights on SOX17 regulation of fertility where epithelial SOX17, via regulation of Indian hedgehog *Ihh* expression, governs uterine epithelial cell proliferation, the ability of the epithelium to support embryo implantation, and the subsequent decidualization of endometrial stroma cells. Cistromic and transcriptomic analyses combined with in vivo editing show that SOX17 regulates *Ihh* expression by acting on an enhancer region 19 kb distal to the *Ihh* promoter. Analysis of the elements binding this enhancer region suggests a general cassette of regulatory elements comprised of transcription factors for the regulation of uterine epithelial gene expression.

## Results

### Mice lacking SOX17 in the uterine epithelium are infertile.
The temporal and spatial expression of SOX17 in the uterine compartments throughout the preimplantation period, Gestation Day (GD) 0.5 to 4.5, were assayed by immunohistochemistry in the pregnant wildtype female mice (Supplementary Fig. 1a). GD 0.5 corresponds to the morning of the presence of the postcoital plug. Staining for SOX17 protein was visible in both luminal epithelium (LE) and glandular epithelium (GE). By GD 1.5, expression of SOX17 in both LE and GE increased and remained constant through the GD 4.5. SOX17-positive cells were also detected in a minority of cells in the stromal compartment (Supplementary Fig. 1a).

The physiological role of SOX17 in the function of the adult uterus was investigated by breeding mice with a conditional allele of *Sox17* (*Sox17^{f/f}* mice)[21] with either *Pgr^{Cre}*[29] or *Ltf^{iCre}*[30] mice to ablate *Sox17* in PGR-positive cells in all compartments of the uterus (*Sox17^{d/d}*) or specifically in the uterine epithelium

(*Sox17^{ed/ed}*), respectively. In addition to being uterine epithelial-specific, gene ablation in the *Ltf^{iCre}* mouse is initiated at puberty[30]. Analysis by quantitative real-time PCR (qRT-PCR) showed reduced *Sox17* mRNA ($P < 0.01$, analysis of variance (ANOVA) with Tukey's post hoc test) in the whole uteri of both *Sox17^{ed/ed}* and *Sox17^{d/d}* female mice as compared with the *Sox17^{f/f}* uteri (Supplementary Fig. 1g). Immunohistochemical analysis showed that SOX17 levels were undetectable in the uterine epithelia of adult female *Sox17^{d/d}* and *Sox17^{ed/ed}* mice (Supplementary Fig. 1b), whereas SOX17 remained expressed in the uterine stromal compartment. These SOX17-positive cells exhibit isolectin B4 reactivity, indicative of an endothelial identity (Supplementary Fig. 1c). Also, no differences in SOX17 expression were detected in uteri between *Sox17^{f/f}* and *Pgr^{cre/+}* mice at GD 3.5 (Supplementary Fig. 1d).

*Sox17^{d/d}* and *Sox17^{ed/ed}* female mice were infertile. Although controls, *Sox17^{f/f}*, mice gave birth to $48.2 \pm 3.6$ pups per female over a 6-month period with an average of $8.0 \pm 0.6$ pups per litter ($n = 4$), neither *Sox17^{ed/ed}* nor *Sox17^{d/d}* female mice gave birth to any offspring during this time. In order to determine the cause of the infertility, *Sox17^{ed/ed}* and *Sox17^{d/d}* female mice were assayed for embryo implantation, uterine stroma decidualization, and epithelial proliferation. Mice were killed at GD5.5 and the uteri were examined for the presence of embryo implantation sites. As shown in Fig. 1a (top panel), whereas *Sox17^{f/f}* showed normal embryo implantation sites, no visible implantation sites were detected in either *Sox17^{ed/ed}* or *Sox17^{d/d}* female mice. Superovulation demonstrated that ovulation in these mice were normal and the implantation defect was owing to the embryos failing to attach to the endometrial epithelium at GD 4.5 (Supplementary Fig. 1e,f). This indicated that the implantation failure was not due to the absence of embryos but rather a failure of these embryos to attach and implant in the uterus.

In addition to the impairment of embryo implantation, the ability of the endometrial stroma cells to undergo an artificially induced decidualization was assayed. The uteri of *Sox17^{f/f}* mice displayed a robust decidual response, as evidenced by the increased size and wet weight of the deciduoma formed in the stimulated left uterine horn ($P < 0.01$, ANOVA with Tukey's post hoc test) (Fig. 1a, bottom panel). However, the uteri of *Sox17^{ed/ed}* and *Sox17^{d/d}* mice failed to form a decidua (Fig. 1a, bottom panel), and the wet weight of the stimulated horn were not different from that of the unstimulated horn ($P > 0.01$, ANOVA with Tukey's post hoc test) (Fig. 1b).

### SOX17 controls epithelial proliferation and differentiation.
During the preimplantation period, the uterus initially proliferates in response to the ovulatory estrogen (E2) surge and as P4 levels rise, the PGR inhibits the E2 induced proliferation. Analysis of uterine epithelial cell proliferation, as measured by Ki67 staining, shows that at GD 3.5 *Sox17^{f/f}* mice exhibited minimal Ki67 staining (H-score; $2.86 \pm 2.00$), whereas both *Sox17^{ed/ed}* (H-score; $217.30 \pm 9.68$) and *Sox17^{d/d}* (H-score; $179.10 \pm 6.03$) uterine epithelium stained positive ($P < 0.05$, ANOVA with Tukey's post hoc test) for Ki67 and thus did not exhibit the inhibition of proliferation during the preimplantation period (Fig. 1c, row 1 and Supplementary Fig. 1h). Interestingly, Ki67 staining in the *Sox17^{ed/ed}* is greater ($P < 0.05$, ANOVA with Tukey's post hoc test) than that of *Sox17^{d/d}*. This change in proliferation is unexpected and may be due to the neonatal verses adult ablation of *Sox17* or the changes in the uterine epithelium due to the focal stratification phenotype.

### SOX17 defines uterine epithelial characteristics.
Analysis of uterine epithelial morphology showed loss of *Sox17* impacted

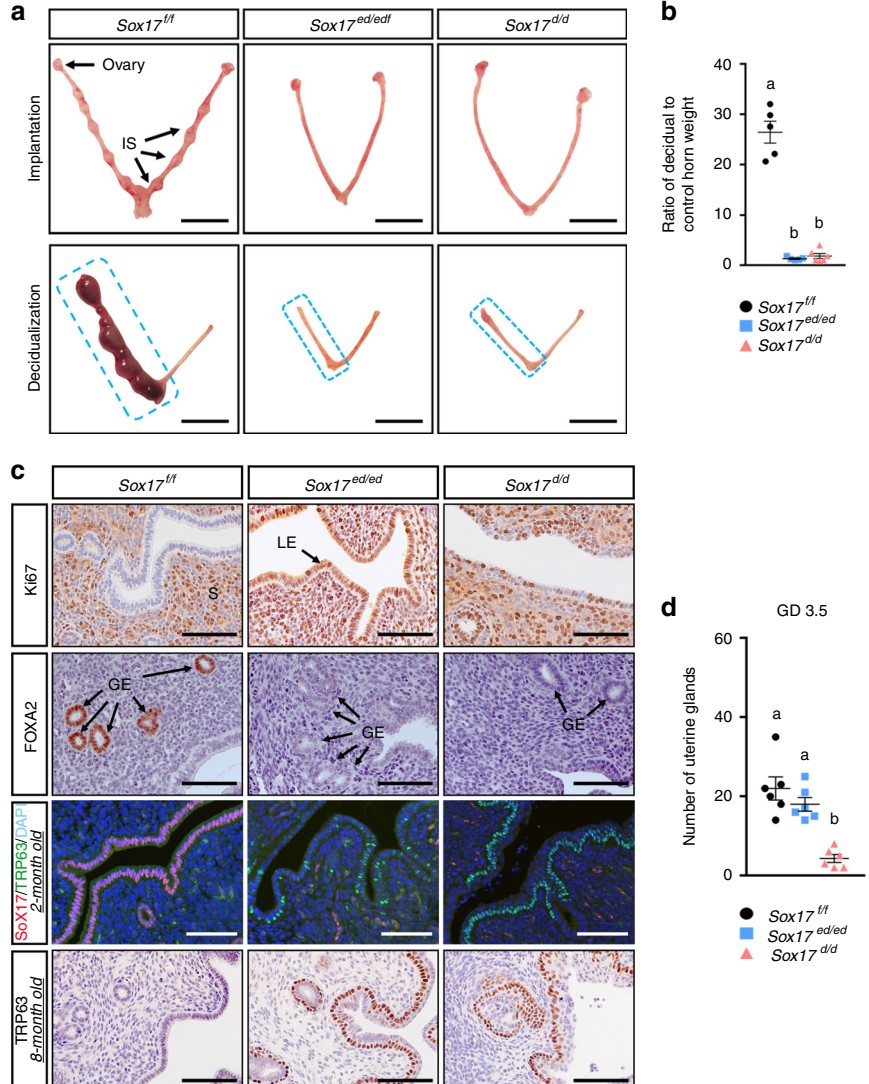

**Fig. 1** Phenotype of conditional deletion of *Sox17* in the adult mouse uterus. **a** Embryo implantation sites (top panel) and artificial deciduoma formation (bottom panel) in *Sox17*$^{f/f}$ (control), *Sox17*$^{ed/ed}$ (*Ltf*$^{iCre}$*Sox17*$^{f/f}$), and *Sox17*$^{d/d}$ (*Pgr*$^{Cre}$*Sox17*$^{f/f}$) female mice ($n = 5$). Scale bars: 10 mm. IS, implantation site. **b** Ratio of decidual to control horn weight in *Sox17*$^{f/f}$ (black dots), *Sox17*$^{ed/ed}$ (blue squares), and *Sox17*$^{d/d}$ (pink triangles) mice ($n = 5$). **c** Immunohistochemical staining of Ki67 (row 1), FOXA2 (row 2), TRP63 (row 3) in the 2-month old uteri of *Sox17*$^{f/f}$, *Sox17*$^{ed/ed}$, and *Sox17*$^{d/d}$ female mice at GD 3.5 ($n = 3$). Row 4 is TRP63 staining in the 8-month old uteri ($n = 3$). Scale bars: 100 μm. S, stroma; LE, luminal epithelium; GE, glandular epithelium. **d** Number of endometrial glands in the 2-month old uteri from *Sox17*$^{f/f}$ (black dots), *Sox17*$^{ed/ed}$ (blue squares), and *Sox17*$^{d/d}$ (pink triangles) mice at GD 3.5 ($n = 3$ mice × 2 sections per mice = 6). Different superscript letters denote significant ($P < 0.05$, ANOVA with Tukey's post hoc test) differences. Data are presented as means ± S.E.M

uterine epithelial cell differentiation. *Sox17*$^{d/d}$ uteri exhibited a reduction in the number of uterine glands as compared with *Sox17*$^{ed/ed}$ and *Sox17*$^{f/f}$ mice (Fig. 1d). We next analyzed the expression of two uterine endometrial gland specific genes, *Foxa2* and *Lif*. FOXA2 transcription factor is a critical regulator of postnatal uterine gland development[31]. Leukemia inhibitory factor (LIF) is a cytokine expressed in the uterine glands and is a critical regulator of embryo implantation[32]. Expression of FOXA2 was undetectable in the GE of both the *Sox17*$^{d/d}$ and *Sox17*$^{ed/ed}$ mouse uteri as compared with *Sox17*$^{f/f}$ mouse uteri (Fig. 1c, row 2). As shown in Supplementary Fig. 1f, both *Foxa2* and *Lif* mRNA were significantly reduced in the *Sox17*$^{ed/ed}$ and *Sox17*$^{d/d}$ mouse uteri as compared to the *Sox17*$^{f/f}$ uteri isolated from GD 3.5 mice. It has been shown that loss of *Foxa2* prior to pubertal onset results in infertility due to loss of uterine glands, whereas loss of *Foxa2* in the adult results in maintenance of the uterine

gland morphology but still loss of fertility due to lack of the GE expression of the cytokine LIF[33]. Therefore, as with FOXA2, *Sox17* is critical for neonatal uterine gland development, and adult uterine gland function.

Previous work has demonstrated that loss of uterine glands results in implantation defects and that the implantation defect can be rescued by administration of recombinant LIF, rLIF, to pregnant mice. In order to determine whether the implantation defect in mice with loss of *Sox17* was solely owing to a glandular defect and loss of LIF, we attempted to rescue the implantation defect by administration of rLIF. In this experiment, mice with deletion of *Foxa2* using the *Pgr*$^{Cre}$ mouse served as a control as this mouse has been shown to have an implantation defect, which could be recused by rLIF[33]. Intraperitoneal injections of mouse rLIF at GD 3.5 did not rescue embryo implantation in *Sox17*$^{ed/ed}$ mice as compared with the positive

controls (*Sox17f/f* mice receiving vehicle; and *Foxa2d/d* mice receiving mouse rLIF) and the negative control (*Sox17ed/ed* mice receiving vehicle) (Supplementary Fig. 1i). This demonstrates that although LIF expression is dependent on SOX17, other pathways regulated by SOX17 in the GE are critical for implantation in mouse uterus.

Analysis of the uterine epithelial morphology showed that in lieu of the simple columnar luminal epithelial layer observed in the *Sox17f/f* uteri, the *Sox17d/d* uteri displayed a second cell layer at 2-month old. This epithelium showed basal cells staining positive for TRP63[34]. The uterine epithelium in the *Sox17ed/ed* female mice did not demonstrate this stratification and TRP63-positive cells were only detected sporadically along the uterine epithelia in 2-month old female mice (Fig. 1c, row 3). Quantification by qRT-PCR showed that *Trp63* mRNA was the highest ($P < 0.05$, ANOVA with Tukey's post hoc test) in *Sox17d/d* uteri, and still high ($P < 0.05$, ANOVA with Tukey's post hoc test) in *Sox17ed/ed* uteri as compared with *Trp63* expression in *Sox17f/f* uteri (Supplementary Fig. 1g). Owing to the different timing of recombination events between *PgrCre* and *LtfiCre*, further morphological analysis and TRP63 staining were conducted in the uteri of 8-month old mice. Both *Sox17ed/ed* and *Sox17d/d* uteri exhibited epithelial stratification with significant increases in the number of TRP63-positive basal cells as compared to *Sox17f/f* uteri (Fig. 1c, row 4). This analysis shows that epithelial SOX17 is critical for the maintenance of the uterine epithelium. This analysis also demonstrated that SOX17 expression in the uterine epithelium is critical for uterine function. As *Sox17ed/ed* and *Sox17d/d* showed a similar phenotype, the following functional analysis of the role of Sox17 was conducted only using the *Sox17ed/ed* mice.

**SOX17 regulates uterine E2 and P4 signaling**. As uterine function is regulated by ovarian steroid hormones[35–38], the effect of *Sox17* ablation on E2 and P4 signaling was investigated. Immunohistochemical analyses of the uteri of GD 3.5, 2-month old *Sox17ed/ed* mice showed increases in total (ESR1) and phosphorylated (pESR1) estrogen receptor in both epithelial and stromal cells (Fig. 2a). Concomitantly, as a downstream target of ESR1, mRNA levels of *Ltf* and *Greb1* were elevated, but *Lif* was downregulated in *Sox17ed/ed* uteri; and no difference in *Lifr* mRNA level was detected between *Sox17f/f* and *Sox17ed/ed* uteri (Fig. 2b). PGR levels are reduced in the epithelial cells, and are lower in the stromal cells owing to lack of induction during the preimplantation period. The downstream target of PGR, *Areg* and *Ihh* were also downregulated in the *Sox17ed/ed* uteri (Fig. 2b).

***Sox17* ablation impairs the Indian hedgehog signaling pathway**. Noting that IHH is a major mediator of PGR signaling in the mouse uterus[3] and significantly reduced in Sox17-ablated uteri, we further investigated the effector downstream of IHH signaling pathways for uterine epithelial–stromal signaling (Fig. 3). Immunohistochemical analyses of the uteri of 2-month old *Sox17ed/ed* mice at GD 3.5 showed decreases in the receptors of IHH, i.e., PTCH1 and PTCH2, mainly in uterine stromal cells, as well as reduction of downstream mediators, COUP-TFII (NR2F2) and HAND2[39] in the stromal cells beneath the uterine epithelium (Fig. 3a). Consequently, elevated phosphorylated-FRS2 (pFRS2) was detected in the uterine epithelium of *Sox17ed/ed* mice (Fig. 3a). qRT-PCR showed a similar pattern of gene expression for of *Ptch1*, *Nr2f2*, *Hand2* as well as *Fgf9* and *Fgf12* in the *Sox17ed/ed* uteri at GD 3.5 (Fig. 3b).

**SOX17 regulates the uterine transcriptome for receptivity**. To further identify the underlying molecular mechanism for the

recurrent implantation failure phenotype observed in the *Sox17ed/ed* mice, an RNA microarray analysis was conducted on whole uterine tissue harvested at GD 3.5 of natural pregnancy. A total of 1595 genes were dysregulated at GD 3.5 with 760 genes upregulated and 835 genes downregulated (Supplementary Data 1). Among dysregulated genes, 125 genes were directly associated with uterine receptivity and implantation as shown in the heatmap (Fig. 4a), including E2-responsive genes (e.g., *Clca3, Fzd10, Greb1, Igfbp5, Wnt4, Ltf, Muc1, Sprr2f*), P4-responsive genes (e.g., *Fosl2, Il6, Msx1, Msx2, Msx3, Osmr, Runx1, Stat3*), LE-specific genes (*Cldn7, Alox12e, Calb1, Cln5, Cnn3, Cobl, Ctsd, Hdc, Irg1, Jam2, Areg*), GE-specific genes (*Calca, Cbs, Foxa2, Ier3, Il6st, Lif, Prss28, Prss29, Spink3*), hedgehog signaling-associated genes (*Ihh, Dhh, Glipr2, Ptch1, Ptch2, Fgf12*), Pan-uterine epithelial-associated genes (e.g., *Btg2, Cd14, Gata1, Ltf, Muc1, Muc16, Mt1, Mt2, Sox17, Sprr2f*) and other uterine receptivity- and implantation-related genes (*Foxo1, Notch1, Notch3*). The expression in the murine endometrium of some of these genes was validated by qRT-PCR (Supplementary Fig. 4). The top-enriched Ingenuity Pathway annotations for the aberrantly regulated gene list (1595 genes) included (1) reproductive system development, function, and abnormalities; (2) cardiovascular system development and function; (3) cancer and dermatological diseases; (4) inflammatory response; and (5) molecular transport (Fig. 4b). The entire list of functional annotations can be found as Supplementary Data 2. Of interest, pathways associated with endometriosis (affected), implantation (deactivated), epithelial cancer (activated), inflammation (activated) as well as transport and secretion of molecules (deactivated) were affected. The pathways altered are involved in developmental cell proliferation and inflammation, which are all important for embryo implantation.

Next, we performed a correlation search to determine which available data sets in the Illumia BaseSpace Correlation Engine database showed similarity to the gene expression profile of *Sox17ed/ed* in the uterus during the window of receptivity. The top two data sets that share similar gene expression patterns with SOX17-regulated transcriptome at GD 3.5 were (1) FOXA2-regulated transcriptome and (2) ARID1A-regulated transcriptome (Fig. 4c). The *Sox17ed/ed* and *Foxa2ed/ed* transcriptomes share 482 affected genes with 212 genes upregulated and 235 genes downregulated (Supplementary Data 3). This finding further supports our hypothesis that the loss of gland epithelial functions in the *Sox17ed/ed* mice is likely through the *Foxa2*-mediated pathway. The *Sox17ed/ed* and *Arid1a ed/ed* transcriptomes shared 344 common dysregulated genes with 106 genes upregulated and 178 genes downregulated (Supplementary Data 4). Although *Arid1a* mRNA level did not changed in the *Sox17* KO microarray, immunohistochemical analyses showed the reduction of ARID1A in the *Sox17ed/ed* uteri as compared with *Sox17f/f* uteri at GD 3.5 (Fig. 4d, row 1), which could be owing to the posttranslational regulation. However, no difference in SOX17 expression was detected between *Arid1a*-ablated and wildtype uteri at GD 3.5 (Fig. 4d, row 2), suggesting a hierarchical role of SOX17 in regulation of ARID1A that is critical for uterine fertility and is remarkably lower in the endometrium from women with endometriosis[40].

**The SOX17 signaling pathway is present in human endometria**. To investigate SOX17 role in human endometrial function, we analyzed its expression pattern in endometrial tissue from women with and without endometriosis at proliferative and secretory phases ($n = 7$ per phase per disease). SOX17 was highly expressed in the endometrial epithelia of women without endometriosis at the proliferative (H-score; $288.00 \pm 8.25$) and secretory phases (H-score $278.57 \pm 9.01$), However, its expression was significantly

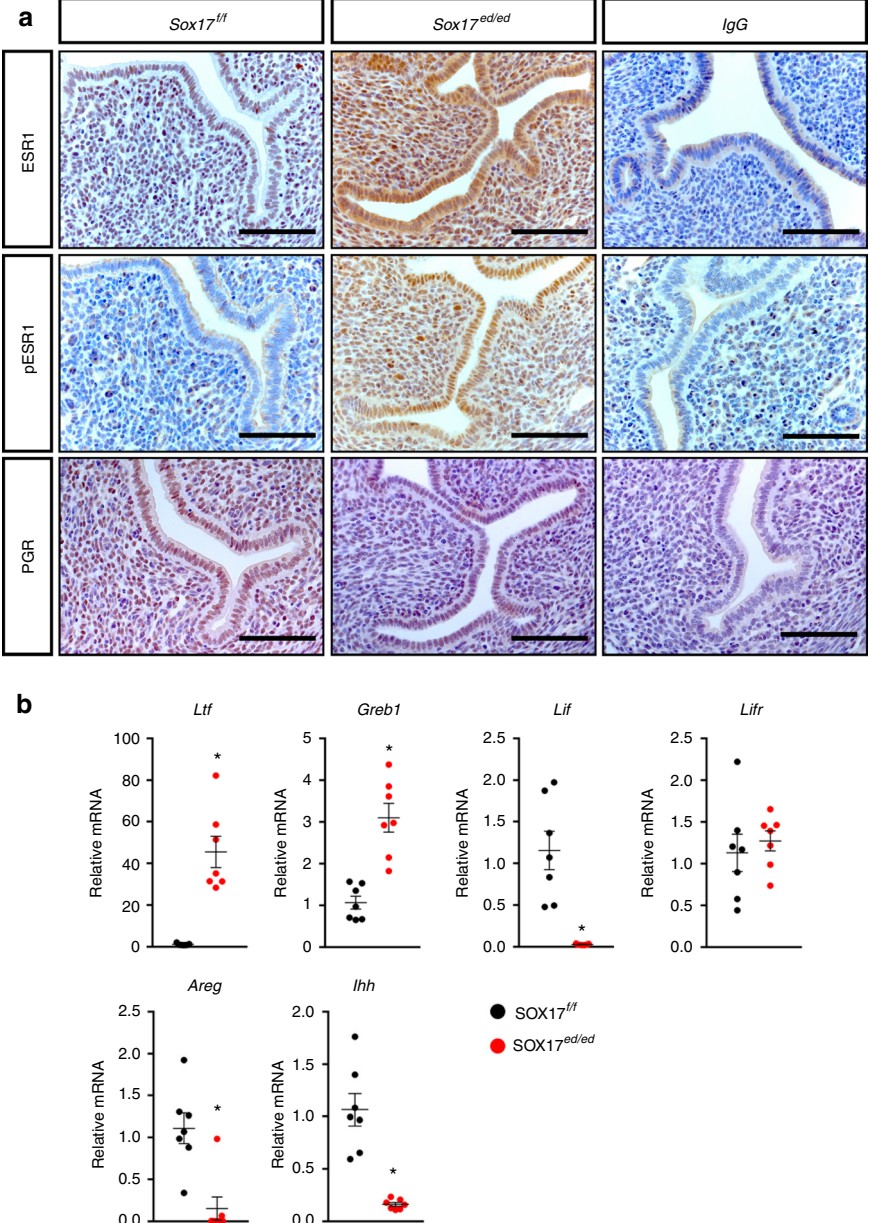

**Fig. 2** Dysregulated estrogen and progesterone signaling in *Sox17*-deficient uterus during the window of receptivity. **a** Immunohistochemical staining of ESR1, phosphorylated ESR1 and PGR in GD 3.5 uteri from *Sox17*f/f and *Sox17*ed/ed mice (*n* = 3). IgG staining as the negative control. Scale bars: 100 μm. **b** Quantification of ESR target genes (*Ltf, Greb1, Lif, Lifr*), as well as PGR target genes (*Areg, Ihh*) in GD 3.5 uteri from *Sox17*f/f (black dots) and *Sox17*ed/ed (red dots) mice (*n* = 7). *$P < 0.05$ (Student's *t* test). Data are presented as means ± S.E.M

decreased in the endometrial epithelia of women with endometriosis at the proliferative (H-score; 119.29 ± 38.83) as well as secretory phases (H-score 48.57 ± 12.94) compared with controls (Fig. 5a, b). This demonstrates that the uterine epithelium expression of SOX17 is conserved between mouse and human, and in the human endometrium SOX17 expression is impacted by endometriosis. Next, the correlation between *SOX17* and *IHH* were examined in the human endometrium. *SOX17* and *IHH* mRNA levels were positively correlated in human endometrium from two independent cohorts, GSE4888 and GSE58144 (Fig. 5c), supporting a conservation of the expression between mouse and human. When comparing genes altered in the *Sox17*-ablated uteri at GD 3.5 with human endometrial receptivity biomarkers[41–43], 31 human orthologs were identified (Fig. 5d). During the window of receptivity, 23 of these genes were downregulated in *Sox17*-

ablated uteri but upregulated in receptive human endometrium; and eight genes were upregulated in *Sox17*-ablated uteri but downregulated in human endometrium, suggesting the role of SOX17 in human and mouse endometrium is conserved.

**SOX17 and major uterine regulators show genome co-occupancy.** As *Sox17* ablation affected uterine epithelial differentiation, it is difficult to discern which gene expression changes are due to direct transcriptional regulation of gene expression or the consequences of epithelial cell differentiation. To define which genes are potentially regulated by SOX17 at the level of transcription, *Sox17*f/f mice were ovariectomized, rested for 2 weeks, and then administered P4. After 6 hours, uterine horns were flash frozen and the tissue was subject to the ChIP-seq assay. The

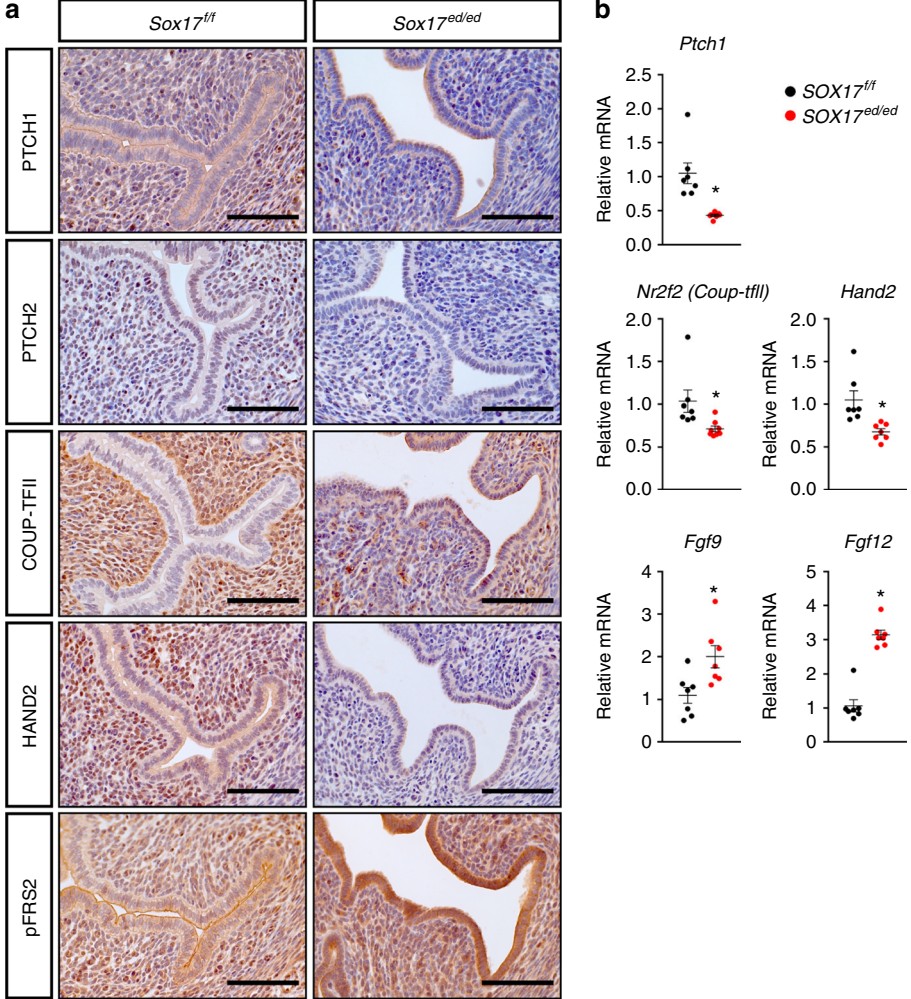

**Fig. 3** Alteration of Indian Hedgehog (IHH) signaling pathways during the window of receptivity. **a** Immunohistochemical staining of IHH receptors PTCH1 and PTCH2, as well as downstream mediator COUP-TFII, HAND2, and phosphorylated FRS in GD 3.5 uteri from *Sox17f/f* and *Sox17ed/ed* mice (*n* = 3). Scale bars: 100 μm. **b** Quantification of IHH signaling-associated genes in GD 3.5 uteri from *Sox17f/f* (black dots) and *Sox17ed/ed* (red dots) mice (*n* = 7). *P < 0.05 (Student's *t* test). Data are presented as means ± S.E.M

SOX17 ChIP-seq demonstrated enriched binding at promoter and 5′ untranslated regions (Supplementary Fig. 3a, b). Homer de novo motif analysis described SOX17 primarily bound at locations with nuclear receptor response elements, in which SOX, GATA families as well as PGR were among top-enriched motifs (Supplementary Fig. 3c). As a result, 11,277 genes were identified as SOX17-bound within 25 kb upstream and/or downstream of the gene boundary (Supplementary Data 5 and 6).

To identify genes potentially regulated by SOX17, differentially expressed genes between *Sox17ed/ed* and control uteri were overlapped with the SOX17-bound gene list. Of the 1595 differentially regulated genes, 807 genes were identified as containing SOX17-binding sites within 25 kb upstream and/or downstream of the gene boundary (Fig. 6a and Supplementary Data 7). Of particular interest, 78 genes have been shown to be associated with uterine receptivity and implantation, 34 of which are uterine epithelial-specific genes (Fig. 6a). Based on the findings that *Sox17ed/ed* and *Foxa2ed/ed* transcriptomes largely overlap and *Sox17* is a target of PGR[5], which is regulated by GATA2 during the early pregnancy[2], we evaluated common binding peaks by comparing SOX17 ChIP-seq dataset with three other independent mouse ChIP-seq data sets (FOXA2[44], PGR[5], and GATA2[2]). This analysis identified 624 common peaks (assigned to 737 genes)

(Fig. 6b). When further overlapping with 1595 differentially regulated genes in *Sox17*-ablated uteri, 92 genes were identified as SOX17 target genes with shared binding sites by FOXA2, PGR, and GATA2. Many of them (*Areg, Btg2, Cnn3, Cobl, Glul, Ihh, Irg1, Lrp2, Pfkfb3, Sox17,* and *Sprr2f*) are well-known uterine epithelial-specific genes, indicating SOX17 may interact with other transcription factors to regulate uterine gene expression.

Interestingly, in vivo SOX17-binding events occurred at putative enhancer 19 kb upstream of *Ihh* gene locus (Fig. 6c). ChIP-seq analysis has demonstrated that FOXA2[44], PGR[5], and GATA2[2] are also bound in this region (Fig. 6c). This region 19 kb upstream of the *Ihh* gene is located at the intragenic region of *Nhej1* gene. *Nhej*−/− mice have been generated and the disruption did not alter the *Ihh19* region. The *Nhej*−/− mice were viable, and fertile[45]. Functional analysis of the role of the *Ihh19* region can be conducted without any concerns that altering the expression of *Nhej* would confound the analysis of the uterine pheyotype. Therefore, to evaluate the significance of these binding sites on uterine receptivity, particularly IHH signaling, we used CRISPR/Cas9 technology to ablate this putative enhancer in vivo (Fig. 6c). Thus, the mouse line (*Ihh19d/d*) was generated with the deletion of 577 bp containing the SOX17 putative enhancer (Fig. 6c and Supplementary Fig. 3d).

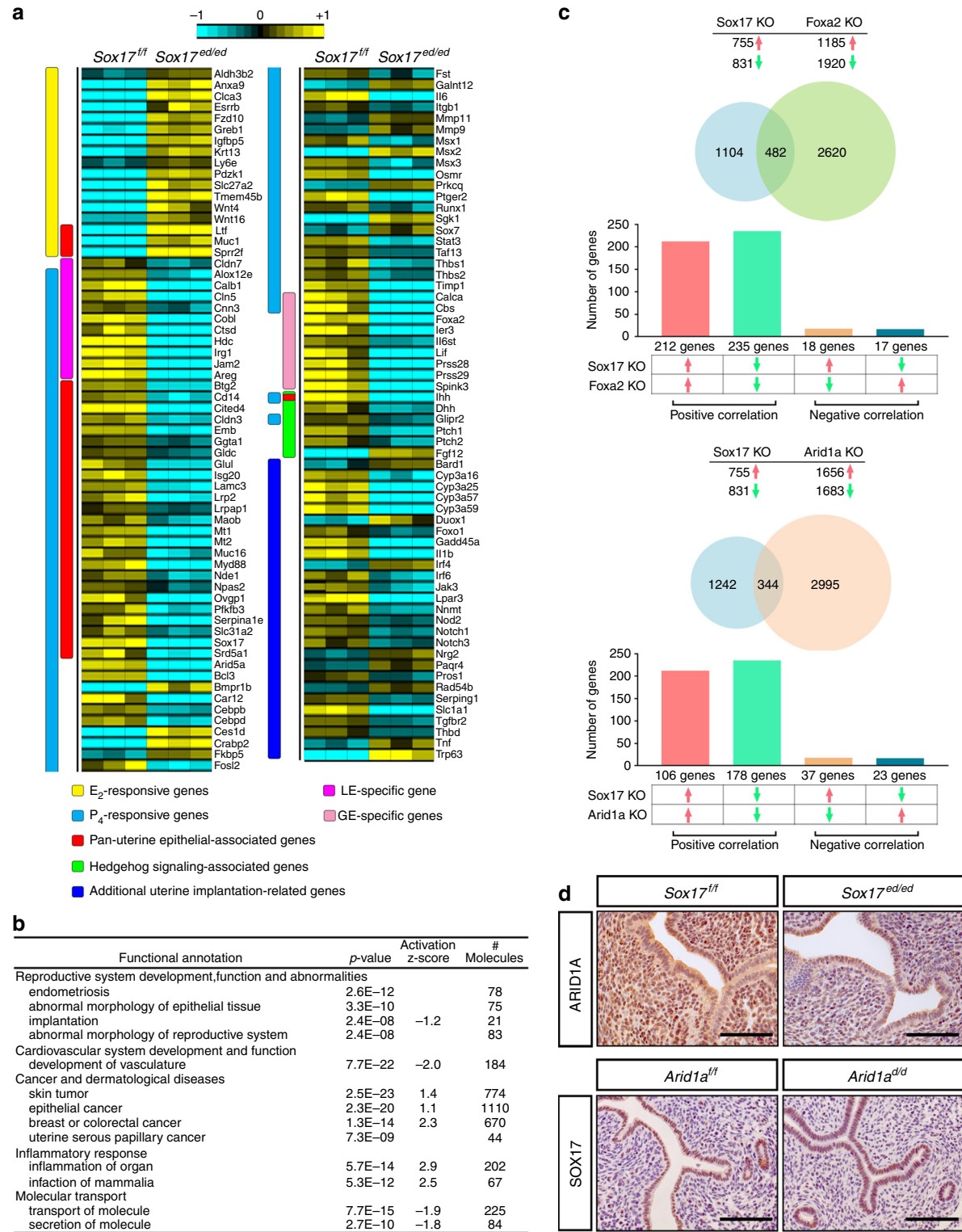

**Fig. 4** SOX17 regulation of uterine transcriptome. **a** Heatmap of uterine function-associated genes differentially regulated between *Sox17f/f* and Sox17*ed/ed* mouse uteri at GD 3.5 generated from microarray analysis. **b** Enrichment of functional annotation in differentially expressed genes between *Sox17f/f* and Sox17*ed/ed* mouse uteri at GD 3.5 by IPA analysis. **c** Overlaps and correlations of *Sox17* KO with *Foxa2* KO or *Arid1a* KO transcriptomes generated by NextBio gene expression profile comparison. **d** Immunohistochemical staining of ARID1A in GD 3.5 uteri from *Sox17f/f* and *Sox17ed/ed* mice (*n* = 3; row 1), as well as SOX17 in GD 3.5 uteri from *Arid1af/f* and *Arid1ad/d* (*PgrCreArid1af/f*) mice (*n* = 3; row 2) Scale bars: 100 μm

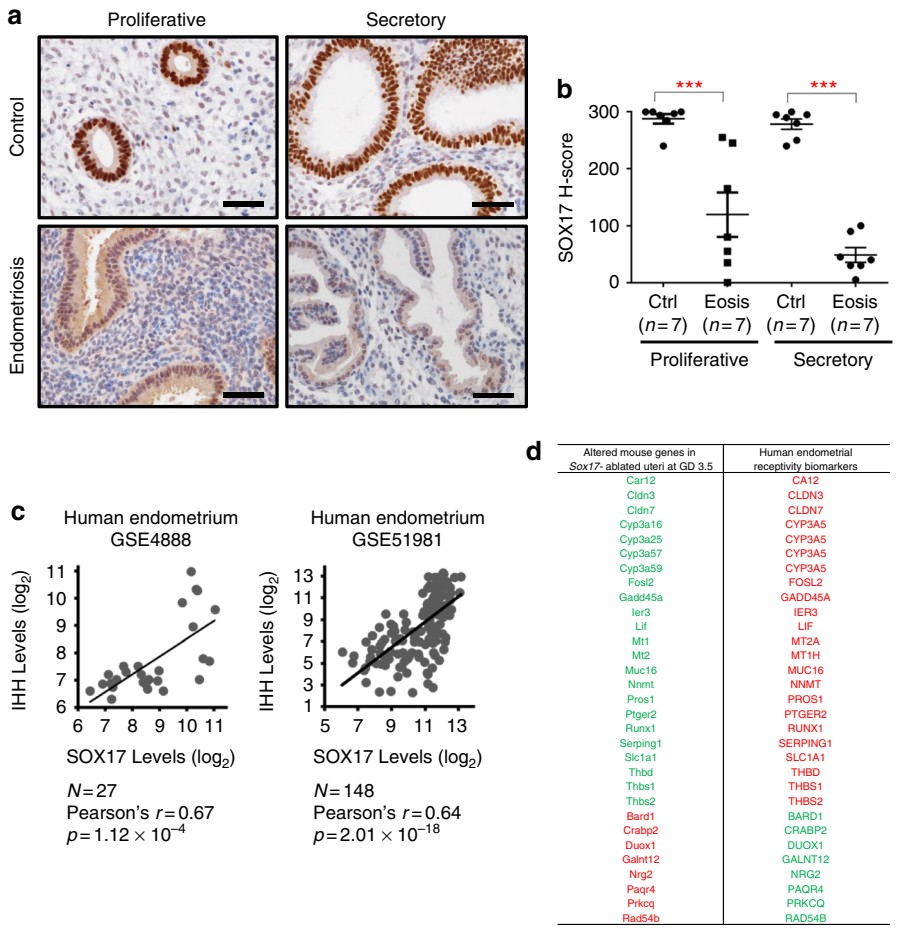

**Fig. 5** A conserved SOX17 pathway for endometrial receptivity in humans. Immunohistochemical staining **a** and H-score quantification **b** of SOX17 in human endometrial section from women with and without endometriosis at proliferative and secretory phase of the menstrual cycle. Scale bars: 50 μm. *$P < 0.001$ (ANOVA with Tukey's post hoc test). Data are presented as means ± S.E.M. **c** Correlation of levels between IHH and SOX17 in human endometrium (GSE4888 and GSE51981). **d** Comparison of genes altered in the *Sox17*-ablated uteri at GD 3.5 with human endometrial receptivity biomarkers. Green font denotes gene downregulation; red font denotes gene upregulation

**An endometrial enhancer for *Ihh* controls uterine receptivity.** Mice bearing the deletion 19 kb upstream of the *Ihh* gene were born at the normal Mendelian frequencies with no obvious skeletal defects as reported in mice with germ line deletion of *Ihh*[46]. This deletion did not impact the expression for *Ihh* that is required for normal development. We then assayed the impact of *Ihh19* deletion on the expression and tissue specificity of *Ihh* in the intestine and uterus of GD 2.5 female mouse. The expression of *Sox17* and *Areg* in the uterus was also assayed (Fig. 7a). Compared with the control (*Ihh*$^{+/+}$, wildtype), *Ihh* mRNA was downregulated ($P < 0.05$) in the *Ihh19*$^{d/d}$ uteri specifically in the uterus with no impact on *Sox17* and *Areg* mRNA levels (Fig. 7a). The expression of *Ihh* was not altered in the intestine (Fig. 7a). Immunohistochemical staining further confirmed the downregulation of IHH in the *Ihh19*$^{d/d}$ uteri at GD 2.5 (Supplementary Fig. 4a). However, the loss of IHH expression was not complete as cells positive for IHH could be identified in the *Ihh19*$^{d/d}$ uteri at GD 2.5 (Supplementary Fig. 4a). As the mice did not show the other non-uterine phenotypes associated with *Ihh* ablation, the deletion of the enhancer of the *Ihh* flanking region impacted was specific to the uterus.

The phenotypic consequences of the deletion of the *Ihh19*-binding region was then assayed based on the ability of the uterus to support pregnancy (including implantation sites), undergo an artificial induced decidual reaction, IHH signaling in the uterus and uterine morphology. *Ihh19*$^{d/d}$ females showed subfertility, as evidenced by the lower number ($P < 0.05$) of pups born per female and pups per litter in 6-month breeding trial as compared with the *Ihh*$^{+/+}$ mice (Fig. 7b). Further, the lower number ($P < 0.05$) of implantation sites at GD 5.5 (Fig. 7c, d) and compromised decidualization (Fig. 7e, f) may account for such subfertility of *Ihh19*$^{d/d}$ females. The decrease in implantation sites was not due to an impact on ovulation or embryo development as there was no difference in the number of embryos flushed from the uterus and oviduct at GD 2.5 between *Ihh19*$^{d/d}$ and control mice at GD 2.5 (Supplementary Fig. 4b). There was no difference in the serum P4 level between these two groups, indicating a normal endocrine regulation of hormone secretion (Supplementary Fig. 4c).

Immunohistochemical analyses showed the expression levels and distribution of uterine epithelial SOX17 and FOXA2 in the uterine GE were not different between *Ihh*$^{+/+}$ and *Ihh19*$^{d/d}$ mice at GD 3.5 (Fig. 7h). Also, there was no TRP63-positive staining for basal cells in the uterine epithelia between them (Fig. 7h). However, analysis of the IHH signaling axis showed that *Ihh19*$^{d/d}$ mice exhibited significant reduction of PTCH1, COUP-TFII, HAND2 in the stromal cells, and increases in pFRS2 mostly in the epithelial cell of uteri at GD

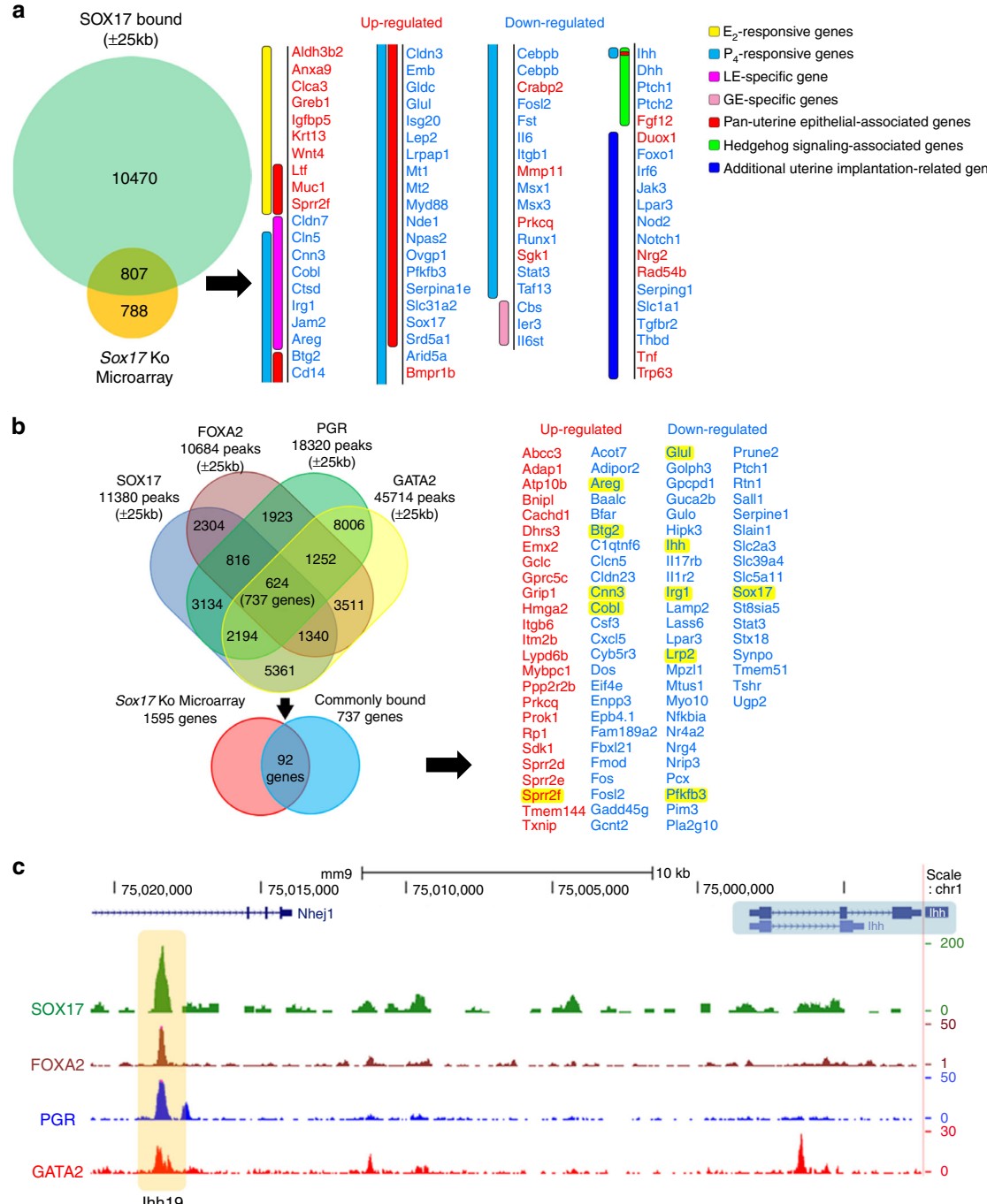

**Fig. 6** Binding and regulation of *Ihh* by analysis of SOX17 uterine cistrome. **a** SOX17 ChIP-seq using *Sox17^{f/f}* mice identified to 11277 genes with binding intervals within 25 kb of the gene boundary. Of these genes, 807 were differentially regulated in the *Sox17* KO microarray at GD 3.5, and 78 genes were directly associated with uterine receptivity and implantation. **b** Analyses of SOX17, FOXA2, PGR, and GATA2 ChIP-seq data sets identified 624 common peaks, which assigned to 737 genes. Of these genes, 92 genes were identified as SOX17 target genes with shared binding sites by FOXA2, PGR, and GATA2. **c** Genome browser tracks of SOX17, FOXA2, PGR, and GATA2 binding at putative enhancers for *Ihh* gene in P₄-treated or GD 3.5 uteri. UCSC Genome Browser views showing the mapped read coverage of SOX17, FOXA2, PGR, and GATA2 ChIP-seq data. The *Ihh* gene locus and the regions targeted by gRNAs for CRISPR/Cas9 deletion in vivo are highlighted in blue and yellow, respectively. Ihh19, putative enhancer 19 kb upstream of *Ihh* gene locus

3.5 (Fig. 7j). Total ESR1 was upregulated mainly in the stromal cells and pESR1 increased in both epithelial and stromal compartments of *Ihh19^{d/d}* uteri as compared with that of *Ihh^{+/+}* mice at GD 3.5 (Fig. 7i). In addition, PGR decreased in epithelial and stromal cells of *Ihh19^{d/d}* uteri as compared with that of *Ihh^{+/+}* mice (Fig. 7i). Although the uterine ablation of

*Ihh* rendered mice completely infertile[3], *Ihh19^{d/d}* mouse exhibited a hypomorphic phenotype of the uterine *Ihh* deletion most likely due to an incomplete ablation of *Ihh* as evidenced by detection ($P < 0.05$) of *Ihh* mRNA in decidua rather than the contralateral horn of *Ihh19^{d/d}* uterus at DD2 (equivalent to GD6.5) (Fig. 7g).

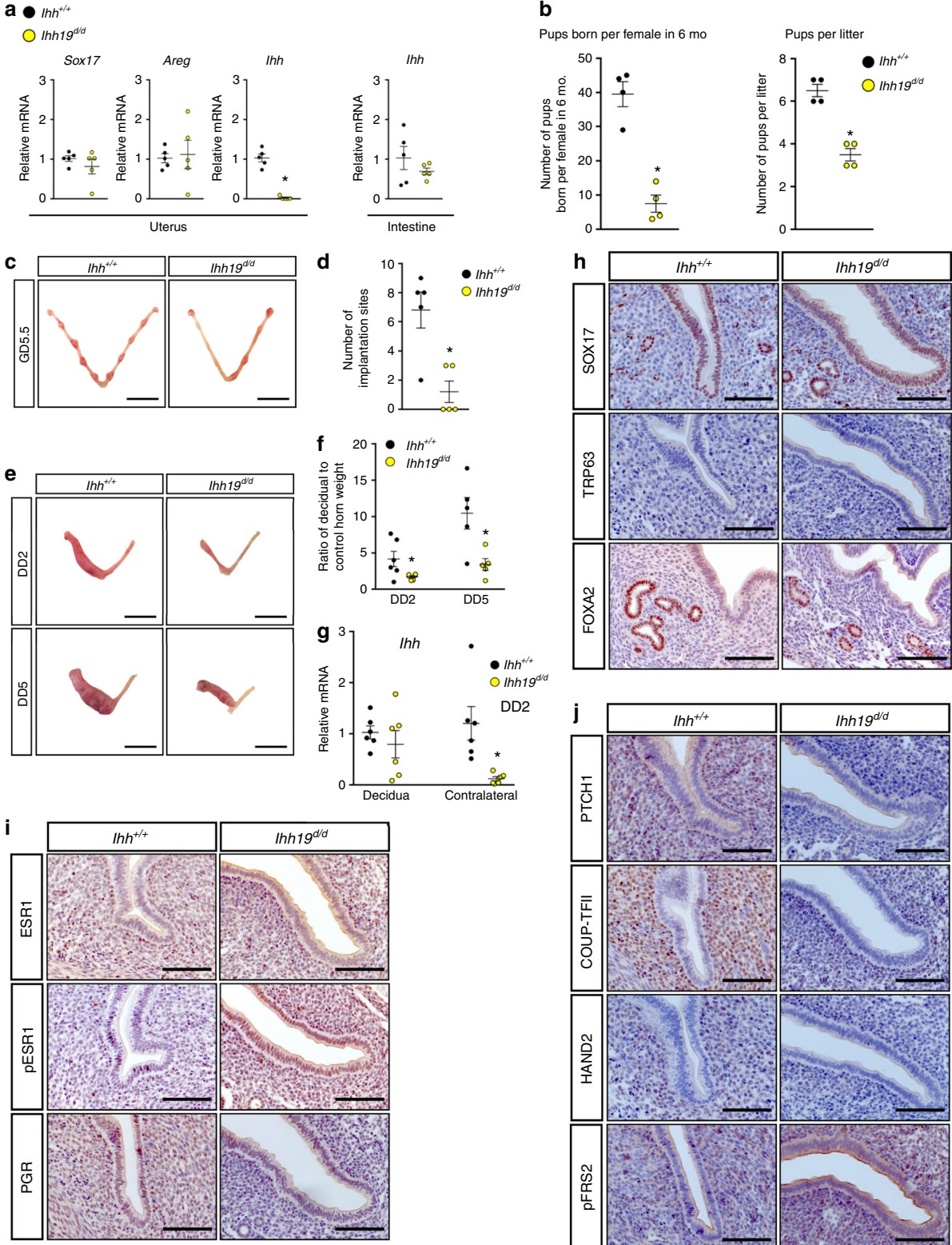

**Fig. 7** The SOX17-binding region 19 kb upstream of *Ihh* gene governs the window of uterine receptivity. **a** Quantification of *Sox17*, *Areg*, *Ihh* gene expression in uteri and/or intestine from *Ihh*[+/+] *and Ihh19*[d/d] female mice at GD2.5 (*n* = 5). **b** Pups born from *Ihh*[+/+] and *Ihh19*[d/d] female mice during 6-month breeding trial (*n* = 4). **c**, **d** Embryo implantation sites (**c**) and quantification (**d**) in uteri from *Ihh*[+/+] and *Ihh19*[d/d] female mice at GD5.5 (*n* = 5). Scale bars: 10 mm. **e**, **f** Artificial deciduoma formation (**e**) and ratio of decidual to control horn weight (**f**) in *Ihh*[+/+] and *Ihh19*[d/d] female mice at DD2 (*n* = 6) and DD5 (*n* = 5). Scale bars: 10 mm. **g** Quantification of *Ihh* gene in decidual and contralateral horn from *Ihh*[+/+] and *Ihh19*[d/d] female mice at DD2 (*n* = 6). **h-j** Immunohistochemical staining of SOX17, TRP63, FOXA2, ESR1, pESR1, PGR, PTCH1, COUP-TFII, HAND2, and pFRS2 in GD 3.5 uteri from *Ihh*[+/+] and *Ihh19*[d/d] mice (*n* = 3). Scale bars: 100 μm. Black dots denote *Ihh*[+/+] and yellow dots denote *Ihh19*[d/d]. *P* < 0.05 (Student's *t* test for **a**, **b**, **d**, and ANOVA with Tukey's post hoc test for **f**, **g**). Data are presented as means ± S.E.M

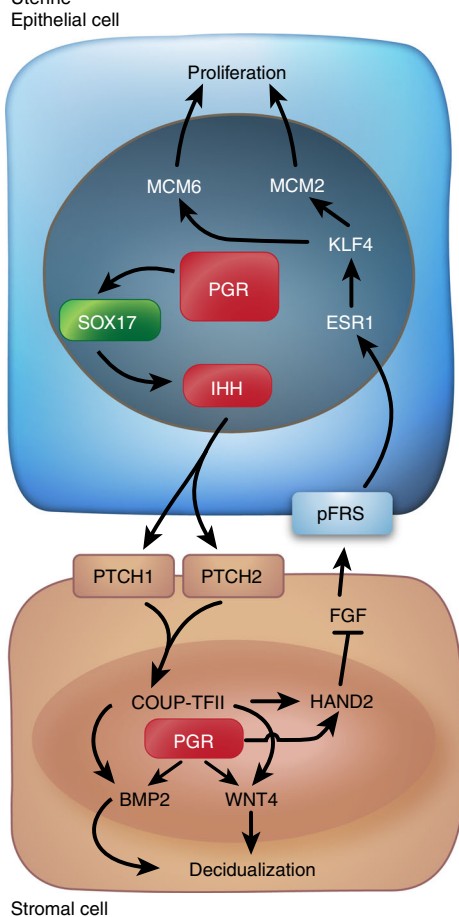

**Fig. 8** SOX17 regulates IHH signaling pathway to govern uterine epithelial–stromal interactions during the window of receptivity

## Discussion

Here we demonstrate that uterine epithelial SOX17 is critical for regulating uterine epithelial cell proliferation, uterine gland development and differentiation, embryo implantation and the ability of the uterus to support pregnancy (Fig. 8). Transcriptomic and cistromic analyses have demonstrated that SOX17 regulates the transcription of uterine epithelial genes. Among the genes regulated by SOX17 is *Ihh*. Ablation of *Sox17* or the SOX17-binding region distal to the *Ihh* gene alters proper epithelial–stromal cross-talk in the uterus required for pregnancy[3,39], in which SOX17 and PGR co-regulated the PGR signaling and IHH signaling. The impaired IHH signal transduction from epithelial to stroma, subsequently affected the stromal PGR via its regulator, the stromal COUP-TFII (Fig. 8). SOX17 expression is conserved in the human endometrium and many of the genes regulated by SOX17 in the mouse are shown to be markers of uterine receptivity in humans.

*Sox17* has been identified as altered in endometrial cancer[26]. Mice generated in this report (i.e., *Sox17^{d/d}* and *Sox17^{ed/ed}*) do not develop endometrial cancer. This indicates that *Sox17* is not a primary driver of endometrial cancer. However, this analysis shows that SOX17 inhibits uterine epithelial cell proliferation. Also, the transcriptomic alterations in response to epithelial *Sox17* ablation overlap with *Arid1a*-ablated transcriptome. ARID1A is highly mutated in endometriosis associated cancers such as ovarian cancer and endometrial cancer[47]. This indicates that the role of *Sox17* in the development of cancer of the uterus

may be in the modification of a critical regulator of endometrial cell homeostasis.

Previously, it has been reported that female mice with *Sox17* haploinsufficiency are subfertile owing to impaired implantation[27]. This was not observed in the *Sox17^{d/+}* or *Sox17^{ed/+}* in this study, in which *Sox17^{d/+}* (*Pgr^{Cre/+}Sox17^{f/+}*) produced an average litter size of 5.3 ± 0.79 pups/litter (*n* = 5 dam), whereas Sox17^{f/+} females gave litters of 5.9 ± 0.89 pups/litter (*n* = 4 dam), suggesting that the decrease in fertility in the *Sox17^{+/−}* mice is likely owing to the non-uterine actions of SOX17. It has also been reported that mice with epithelial deletion of *Sox17* using the *Sprr2f^{Cre}* mice are infertile with normal adenogenesis[28]. Our findings further show that ablating *Sox17* in the uterine epithelium by the *Ltf^{iCre}* results in complete abolition of embryo implantation, which not only agrees with the previous report but also provides a more detailed functional timeline of epithelial Sox17 on fertility. The *Sprr2f^{Cre}*-mediated *Sox17* knockout model also show the presence of glands in the uterus, which differs from that of the *Pgr^{Cre}*-mediated *Sox17* ablation model[28]. It was speculated that the persistence of glands was due to an involvement of SOX17-expressing stromal cells. Our results identified the SOX17-expressing cells in the stromal compartment as endothelial cells. Moreover, deleting *Foxa2* in uterine epithelium after puberty also allows establishment of glands despite of sterility[33]. Therefore, the presence of glands in both *Sprr2f^{Cre}* and *Ltf^{iCre}* models are likely owing to the timing of *Sox17* ablation. In both models, gene recombination is delayed until after puberty[30,33,48].

The role of *Sox17* in the mouse uterus is twofold. First, it regulates uterine gland development and differentiation. Second, it regulates the expression of key genes regulating uterine fertility. The loss of epithelial *Sox17* postnatally reduced the number of glands and caused the epithelial stratification in the 2-month old mouse uteri. Postpubertal ablation of *Sox17* resulted in a delay in epithelial stratification and no glandular reduction in the 8-month old uteri. However, uterine gland function was altered as evidenced by a decrease in *Lif* expression. This suggests that, like *Foxa2*, *Sox17* regulates postnatal adenogenesis and once the uterine glands are developed they regulate gland function. Furthermore, the significant reduction of FOXA2 in the GE of *Sox17*-ablated uteri at GD 3.5, and no binding of SOX17 to the *Foxa2* gene locus suggest an indirect regulation of SOX17 on *Foxa2* gene expression. Notably, the loss of LIF in this model was not the only cause of the infertility as administration of recombinant LIF did not rescue the implantation phenotype. This observation suggests that additional pathways parallel to the FOXA2-LIF axis are also required for postpubertal gland functions.

The epithelial stratification with the expression of basal cell marker TRP63 in *Sox17* KO is similar to the uterine phenotypes seen after ablation of β-catenin[49], *Wnt4*[50], or *Gata2*[2] and conditional uterine activation of smoothened[51]. This places SOX17 as a regulator of uterine epithelial integrity along with P4, WNT, and hedgehog signaling. SOX17 is reported to inhibit WNT signaling[52]. However, in this case SOX17 loss phenocopies WNT4/β-catenin loss, indicating that its action in the uterine epithelium is not to inhibit WNT signaling but work in concert with these signaling pathways to regulate uterine epithelial differentiation.

The cause of infertility in the *Sox17^{ed/ed}* mouse is due to the loss of two major signaling pathways. Both LIF and IHH signaling are altered in this mouse model. LIF is produced by the GE and acts on its receptors in the LE to regulate embryo invasion and stroma decidualization[32]. IHH initiates the epithelial–stromal communication network, which is required for cessation of uterine epithelial cell proliferation, embryo invasion, and decidualization[3]. However, the mechanism by which SOX17 alters these pathways is confounded by its action to regulate epithelial

and glandular differentiation. Similar transcriptomic alteration between $Sox17^{ed/ed}$ and $Foxa2^{ed/ed}$ further suggest the essential role of SOX17 in adenogenesis and maintenance of functional glands in the uterus. Transcriptomic, cistromic, and genome editing analyses indicate that the role SOX17 plays in regulating uterine function may not be secondary to alterations in uterine epithelial differentiation, but may be due to direct regulation of genes critical for uterine fertility. Analysis of the binding sites for SOX17 demonstrated 737 genes that share the common binding sites (624 peaks) for PGR, GATA2, and FOXA2. Among those genes, 92 were identified in response to SOX17 regulation via the shared uterine-specific enhancers, i.e., non-coding regions harboring *cis*-regulatory elements that are bound by transcription factors and may define cellular identity[53,54]. The putative enhancer 19 kb upstream to the *Ihh* gene was identified based on SOX17, FOXA2, PGR, and GATA2 ChIP-seq data from uterine tissue. Thus, *Ihh19* enhancer was further deleted in vivo using CRISPR/Cas9 technology and validated by sequencing. The mice were subfertile with impaired IHH signaling only in uterine tissue, confirming the uterine distal enhancer bound by SOX17 that regulates *Ihh* gene. The analysis of this enhancer region has identified a cassette of transcription factor binding sites that may be critical for uterine epithelial gene expression. It is easy to speculate that some combinations of these transcription factors would allow the differentiation of progenitor cell to achieve a uterine epithelial cell phenotype. Uterine epithelial cells are difficult to culture in vitro, therefore understanding regulation of and by these transcription factors may lead to the development of better in vitro models for uterine epithelial function.

## Methods

**Human endometrial samples**. This study was carried out in accordance with federal regulations governing human subjects research. All procedures were approved by the following ethics committees: Institutional Review Board/Committee-A (IRB) of Greenville Health System under IRB file # Pro00000993 and Pro00013885 and the University of Capel Hill at North Carolina IRB under file #: 05-1757. Informed consent was obtained from all patients before their participation in this study. To examine SOX17 expression patterns of eutopic endometrium between women with and without endometriosis (control vs. endometriosis group), 28 samples in total were used from proliferative and secretory phases ($n = 7$ per phase per group).

**Animals**. All animal studies were conducted in accordance with National Institute of Environmental Health Sciences (NIEHS) Animal Care and Use Committee guidelines and in compliance with NIEHS-approved animal protocol. Mice carrying the $Sox17^{f/f}$ allele (stock no: 007686, the Jackson Laboratory, Bar Harbor, ME) were bred to $Ltf^{icre/+}$ or $Pgr^{cre/+}$ mice to generate $Ltf^{icre/+}Sox17^{f/f}$ or $Pgr^{cre/+}Sox17^{f/f}$ mice, respectively.

**CRISPR/Cas9-mediated deletion of the enhancer region in vivo**. CRISPR/Cas9 single-guide RNAs targeting *Ihh19* enhancer region were identified using MIT CRISPR Design tool (http://crispr.mit.edu)[55–57], i.e., GGTACAGACTGGAGCCCTTA, and TTATAGATAGCAGGCACTAT. Each sgRNAs were cloned into the pDR274 plasmid vector, and in vitro transcribed using the MEGAshortscript T7 kit (Life Technologies). Cas9 mRNA was in vitro synthesized. Cas9 mRNA (50 ng $\mu l^{-1}$) and sgRNAs (25 ng $\mu l^{-1}$ for each) were microinjected into the cytoplasm of fertilized eggs of superovulated C57BL/6 J female mice (JAX) and implanted into oviducts of pseudopregnant fosters. Founder mice were bred with wildtype C57BL/6 J mice to obtain F1 heterozygous ($Ihh19^{d/+}$) mice. F1 mice with an identical genotype were interbred to generate F2 homozygous ($Ihh19^{d/d}$) mice. All mice were genotyped by PCR amplification of genomic DNA isolated from the tip of tail, followed by Sanger sequencing. Large deletions were identified by serial PCR genotyping using primers that were designed to amplify ~500 bp encompassing the target sequence.

**Animal fertility assay**. Fertility was assessed by mating 8-week old (1) $Ltf^{icre/+}Sox17^{f/f}$ or $Pgr^{cre/+}Sox17^{f/f}$ females; and (2) $Ihh19^{+/+}$ or $Ihh19^{d/d}$ females with wildtype C57BL/6 J males for 6 months. The number of litters and pups were recorded.

**Embryo implantation assessment**. Eight-week old female $Sox17^{f/f}$, $Ltf^{icre/+}Sox17^{f/f}$ and $Pgr^{cre/+}Sox17^{f/f}$ mice, as well as female $Ihh19^{+/+}$ and $Ihh19^{d/d}$ mice were

mated with C57BL/6 J intact male mice, separately. The presence of the copulatory plug was recorded as GD 0.5. Mice were killed by injection of sodium pentobarbital (Fatal Plus; Vortech Pharmaceuticals, Dearborn, MI) in the morning, 5 days (i.e., GD 5.5) after the presence of the coital plug, to determine the number of implantation sites.

**Artificial decidualization**. To assess the ability of the stroma to undergo differentiation and proliferation independent of embryo attachment, 6-week old female mice were ovariectomized and treated with exogenous hormones to mimic pregnancy before applying a manual stimulus to a single uterine horn (protocol outlined previously by Finn and Martin, 1972)[58]. After ovariectomy and 2 weeks of rest to eliminate endogenous ovarian steroids, mice were administered daily E2 (100 ng per mouse) injections for 3 days. After 2 days of rest, mice were treated with E2 (6.7 ng per mouse) and P4 (1 mg per mouse) for 3 days. On the third day (DD0), mice were administered a single injection of 0.05 mL of sesame oil to the right uterine horn. Mice were administered E2 and P4 for 5 more days and killed on the fifth day (DD5). Uterine wet weights for the stimulated and control horns were recorded. Weight ratios were calculated by dividing stimulated horn weight by unstimulated horn weight.

**Superovulation assay**. Superovulation was induced in female mice by i.p. injection of five international units (IU) of pregnant mare's serum gonadotropin (EMD Millipore, Billerica, MA), followed by 5 IU of human chorionic gonadotropin (Pregnyl, Merck & Co., Inc., Whitehouse Station, NJ) 48 h later and placed with wildtype C57BL/6 J male mice. The mice were killed 24 h later by cervical dislocation while under the anesthetic, Avertin (2.2-tribromoethyl alcohol, Sigma-Aldrich, St. Louis, MO) and oocytes were flushed from the oviducts and counted.

**LIF rescue assay**. The attempt to rescue the implantation defect in the $Sox17^{ed/ed}$ mice was performed using the protocol described by Kelleher et al.[33]. In brief, female mice received two intraperitoneal (i.p.) injections (one at 1000 h and one at 1800h) of vehicle or 10 µg recombinant mouse LIF on GD 3.5, and embryo implantation sites were evaluated on GD 5.5.

**Immunohistochemistry and dual immunofluorescence**. Tissues were fixed in 4% paraformaldehyde and embedded in paraffin wax. Embedded tissues were sectioned at 5 µm and baked 1 hour at 60 °C. Upon cooling, slides were dewaxed using Citrisolv clearing agent (catalog no. 22-143-975, Thermo Fisher, Waltham, MA, USA) in a decreasing gradient of pure ethanol. For hematoxylin and eosin (H&E) staining, tissues were adequately stained with H&E and were then dehydrated before coverslips were applied. For immunohistochemistry, antigen retrieval was performed according to manufacturer's instructions (Vector Labs Antigen Unmasking Solution H-3300). Endogenous peroxide was blocked using 3% hydrogen peroxide diluted in methanol. The tissue was blocked with 5–10% normal donkey serum before application of primary antibody overnight at 4 °C. Secondary antibody was diluted in 1% bovine serum albumin at a concentration of 1:200 when required. The ABC reagent was applied to tissues according to manufacturer's instructions (Vestor Labs ABC PK-6100). Signal was developed using Vector Labs DAB ImmPACT staining according to manufacturer's instructions (Vector Labs SK-4105). Tissue was counterstained with hematoxylin and dehydrated before affixing coverslips. A semiquantitative grading system (H-score) was used to compare the immunohistochemical staining intensities[59]. The H-score was calculated using the following equation: H-score $= \sum Pi$ (i), where $i =$ intensity of staining with a value of 1, 2, or 3 (weak, moderate or strong, respectively) and $Pi$ is the percentage of stained cells for each intensity, varying from 0 to 100%. For dual immunofluorescence, antigen retrieval was performed and endogenous peroxide was blocked as aforementioned. After the tissue was blocked with 5–10% normal donkey serum for 1 hour, two primary antibodies from different species were applied simultaneously overnight at 4 °C. Sections probed with primary antibodies were incubated with two respective secondary antibodies (Alex 488 and 568) at dilutions of 1:200 for 1 hour at room temperature and then rinsed in phosphate-buffered saline and overlaid with Prolong Gold Antifade with 4′,6-diamidino-2-phenylindole. Slides were stored at 4 °C in the dark before microscopic analysis. Information for all antibodies is provided in Supplementary Table 1.

**RNA isolation**. Frozen tissue was homogenized in TRIzol reagent (Thermo Fisher). Isolation of RNA was performed using chloroform and precipitated using isopropanol with resuspension in water. For RNA prepared for microarray, TRIzol reagent was utilized followed by the aqueous phase isolation using 1-Bromo-3-chloropropane and a second aqueous phase using chloroform. Pure ethanol was applied to the aqueous layer, and the total solution was administered to the Qiagen RNEasy RNA mini prep kit column. The column was washed and the RNA was isolated using manufacturer's instructions (Qiagen, Valenica, CA).

**Reverse transcriptase PCR and QRT-PCR**. Reverse transcription of RNA into cDNA was performed using MMLV reverse transcriptase (Thermo Fisher) according to manufacturer's instructions. QRT-PCR was performed using Taqman Master Mix (Life Technologies) and Taqman probes (Applied Biosystems). Delta

delta Ct values were calculated using 18 S control amplification results to acquire relative mRNA levels per sample. Information for all primers is provided in Supplementary Table 2.

**Microarray analysis**. RNA quality was assessed using the Agilent 2100 Bioanalyzer (Agilent Technologies). Microarray analysis was performed by the Genomic and RNA Profiling Core at Baylor College of Medicine. After library amplification and labeling, individual cDNA samples were hybridized to the Agilent G3 Mouse GE 8 × 60 k array according to manufacturer's instructions (Agilent Technologies). Array data were analyzed using Bioconductor with quantile normalization. Genes with an unadjusted $p$ value < 0.01, and an absolute fold change ≥ 1.4 were identified as differentially regulated.

**Chromatin immunoprecipitation and sequencing**. Whole uterine tissue was flash frozen and sent to Active Motif company for Factor-Path chromatin immuno-precipitation and sequencing (ChIP-seq) analysis. The tissue was fixed and then promptly sheared into small fragments before immunoprecipitation with the SOX17 antibody (AF1924; R&D Systems). SOX17-bound DNA was purified and amplified to generate a library and sequenced using Illumina's Genome Analyzer 2. The raw ChIP-Seq reads (36 nt, single-end) were processed and aligned to the mouse reference genome (mm9; Genome Reference Consortium Mouse Build 37 from July 2007) by Active Motif. SOX17-bound intervals were identified using Model-based analysis of ChIP-Seq (MACS 1.3.7.1) and mapped to nearby genes within 25 Kb by developing a Perl script.

**Data analysis**. The differentially expressed genes generated from microarray data were analyzed using Ingenuity Pathway Analysis software (IPA, http://www.ingenuity.com) and Database for Annotation, Visualization, and Integrated Discovery (DAVID, http://david.ncifcrf.gov/). The similar transcriptomes to the $Sox17^{ed/ed}$ gene expression profile were identified by searching available data sets in the NextBio database (https://www.nextbio.com/). The ChIP-Seq data were analyzed using the Cistrome software (http://cistrome.org/ap/). The GraphPad Prism software was implemented for one-way ANOVA, multiple comparison test, and Student's $t$ test analyses for qRT-PCR and decidual wet weights. Hormone response elements were identified using HOMER de novo motif analysis (http://homer.salk.edu/homer/). Hierarchal clustering heatmaps were generated using Partek Genomics Suite 6.6 software.

## Data availability

The data that support the findings of this study are available from the article and Supplementary Information files, or from the corresponding author upon request. Raw FASQ data files (GSE118264) and Genome-wide sequencing data (GSE118327) are deposited in the NCBI Gene Expression Omnibus database under SuperSeries accession number GSE118328.

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

## Acknowledgements

We thank Dr. Paul Labhart from Active Motif, Dr. Nyssa R. Adams, Mita Ray, Linwood Koonce, the NIH library manuscript editing service, the Genetically Engineered Mouse, and the Microarray Cores at BCM, and the Knockout Mouse Core and the Comparative Medicine Branch at NIEHS for assistance in this study. We also thank Dr. Sophia Y. Tsai, and Sylvia C. Hewitt for critical comments. This work was supported by the Intramural Research Program of the National Institute of Environmental Health Sciences: Project Z1AES103311-01 (F.J.D.), and NIH/NICHD: R01: HD-042311 grant (J.P.L.)

## Author contributions

X.W., X.L., J.P.L. and F.J.D designed the experiment. X.W. and X.L. conducted experiments including: animal fertility assay, embryo implantation assessment, artificial decidualization assay, immunohistochemistry, immunofluorescence, qRT-PCR, Microarray, and ChIP-seq analyses. X.L. and F.J.D. established and performed initial characterization of the *Ihh19* deletion mouse line. J.J. and T.H.K. conducted experiments related to the *Arid1a* deletion mouse model and human endometrial tissues. S.L.Y. and B.A.L. collected human endometrial specimens. X.W., T.W., S.W. and R.B.L. each performed a subset of data analysis. X.W., J.J. and F.J.D. wrote the manuscript.
