## [Peer Review File · Nature Communications]

Reviewers' comments:

Reviewer #1 (Remarks to the Author):

In the manuscript "SOX17 regulates uterine epithelial-stromal crosstalk acting via a distal enhancer upstream of *Ihh*", authors have used genetic mouse models and sequencing analysis to identify the function of transcription factor Sox17 in uterine receptivity. Authors claim that Sox17 regulates uterine epithelial-stromal crosstalk through a distal enhancer upstream of *Ihh*.

Major concerns:

- 1) This reviewer's biggest concern is that authors do not make a very clear distinction, if the function of Sox17 is essential in luminal epithelium or glandular epithelium or both. Most of the immunohistochemical data focuses on the luminal epithelium as uterine receptivity is a function of luminal epithelium. While this reviewer agrees that Sox17 is essential for endometrial receptivity, this reviewer feels that the role of glandular epithelium has been overlooked. Given the author's data there are three pieces of evidence that point out that this is at least as much a gland phenotype as luminal epithelium receptivity phenotype:
 - a) The Sox17 array overlapped with the Foxa2 array – evidence that this is a gland phenotype not just a general epithelium phenotype.
 - b) The Sox17 binding peak was also bound by FOXA2 and since FOXA2 is only in the glandular epithelium this would suggest that the gene set essential for receptivity and the enhancer region identified by the authors is important in glandular epithelium as opposed to the whole epithelium.
 - c) When Sox17 binding enhancer is deleted upstream of *Ihh*, FoxA2 expression is intact in the glands and there is a partial fertility phenotype.

In order to address the separate function of Sox17 in glandular epithelium vs luminal epithelium and to also compare further this phenotype with the FoxA2 deletion phenotype, authors should inject LIF at the appropriate time and determine if LIF supplementation alone will rescue part of the phenotype. If in case there is complete rescue (like in the case of *LtfCre*;FoxA2 deletion) then the phenotype is completely a gland phenotype. If there is a partial rescue that will support Sox17 having a role both in glandular epithelium and luminal epithelium.

- 2) Since Sox17 is a regulator of PGR action and Sox17 is also target of Pgr – how does using PGRcre by itself affect Sox17 expression? Is the Sox17 d/d phenotype exaggerated by loss of one copy of PGR? The authors should comment on this in the manuscript.
- 3) The authors choose to analyze expression of Sox17 in a pseudopregnant uterus. The rationale is unclear to this reviewer as to why a normal pregnant uterus was not evaluated.
- 4) In the Sox17 deletion mutant – authors suggest that loss of Sox17 leads to reduction of PGR in the stromal cells – is this a reduction or since PGR is induced in the stromal cells at this time in pregnancy, there is lack of induction of PGR in the stroma?

- 5) Is Arid1a expression down regulated in the Sox17 dataset?
- 6) Since the microarray matched up to both FoxA2 and Arid1a, this phenotype is likely a common gene expression signature in the absence of receptivity or pregnancy. This should be emphasized more in the manuscript and perhaps in the title.
- 7) This reviewer is confused on why the CHIP was performed with Ovx and P4. Is there PGR in the uterine epithelium in mice that have been ovariectomized and kept for two weeks? If not, then how will the P4 activate the PGR specific gene set?
- 8) In order to confirm loss of Ihh specifically in the uterus, the authors need to show Ihh protein expression loss by IHC or IF in the Ihh enhancer deleted uteri.
- 9) The authors point to a role of Sox17 in epithelial – stromal communication – but the data is very weak in support of this claim and this is concerning as this is also a part of the title of the manuscript.
- 10) Why is pESR1 staining the luminal epithelial cell membrane in control in Fig. 2a?
- 11) Fig 2a, sections, for Sox17 deletion used for ESR1 and pESR1 staining look very different from PGR staining? Is there a variable phenotype where luminal epithelium is sometimes pseudostratified and sometimes single layer columnar?

Reviewer #2 (Remarks to the Author):

An earlier publication using heterogeneous Sox17 KO mice revealed that critical role of Sox17 in embryo implantation (PMID 27053385). Another previous study revealed crucial role of Sox17 in uterine gland formation and fertility (PMID 27102016). It also demonstrated that Pgr-Cre/Sox17-loxP (uterus-specific Sox17 KO, Sox17d/d) mice and Sprr2f-Cre/Sox17-loxP (uterine epithelium-specific Sox17 KO) mice are infertile and FoxA2 is a target of uterine Sox17, indicating epithelial Sox17 is essential for embryo implantation. Based on these studies, the current study by Wang X et al. confirmed that implantation is impaired in mice with epithelium-specific deletion of Sox17 (Ltf-iCre/Sox17-loxP, Sox17ed/ed). Importantly, this study newly found that uterine Sox17 regulates the implantation-related factors and pathways such as FoxA2-LIF pathway and Ihh signaling pathway. Notably, this study showed that mice with deletion of Sox17-binding site in Ihh locus are subfertile with reduced uterine induction of its downstream genes, indicating Sox17 directly regulates Ihh in the uterus.

Although the current study includes some novel findings, it is unclear what is conceptual advance provided by this study. All the targets which authors provided are well-known factors and pathways in the process of implantation. Which pathway driven by Sox17 is most critical for implantation? Can LIF supplementation recover implantation failure in Sox17ed/ed mice? I think contribution of stromal Sox17 is not negligible, so the authors should examine the role of stromal Sox17 in vivo.

Specific comments are as follows:

1. In Line 328, the authors described that epithelial deletion of Sox17 using Sprr2fCre mice are fertile. According to the reference (PMID 27102016), their wrong description might be due to their misunderstanding. Mice with Sprr2fCre-induce epithelial deletion were infertile, but did not impair gland formation. Reproductive phenotypes in this reference are similar to those shown by the current study.
2. In Figure 1c, Ki67 staining in Sox17d/d mice look almost normal, while epithelium-specific Sox17 deletion promotes epithelial proliferation in Sox17ed/ed mice. How does stromal and epithelial Sox17 control cell proliferation and differentiation? Does stromal deletion of Sox17 normalize epithelial differentiation?
3. The authors previously showed importance of P4-Pgr-Ihh signaling in uterine receptivity. The present study showed that deletion of Sox17 reduces the expression of Pgr and Ihh. Is the downregulation of Pgr a major cause of dysregulation of Pgr signaling in Sox17ed/ed mice? In Figure 2a, how is stromal expression of Pgr downregulated in Sox17ed/ed mice?
4. It is difficult to understand the epithelial-stromal interaction through Sox17 and other implantation-related molecules, because so many implantation-related molecules were examined with no distinction between stromal and epithelial specific genes in this paper. For example, Supplemental figure 4 contains both epithelial and stromal specific genes. I struggled to understand the relationship among these key factors for implantation, especially in the latter half of manuscript. Please edit the text and make it understandable. Schematic figures might be useful.
5. In Figure 5a, are the endometriosis samples endometriotic lesions, or eutopic endometria in the patient with endometriosis? Are the endometria of proliferative and secretory phase obtained from women with or without endometriosis? Background of human samples should be described in detail. how many human samples were investigated in each group? Quantification of immunostaining and statistical analysis should be recommended.
6. The figure and the legend of Figure 6c needs to be revised to make it understandable.
7. In Ihh19d/d mice, ovarian function including ovulation, fertilization and hormone production should be assessed.

Response to Reviewers Comments:

We appreciate the reviewers' comments and have added new significant data, analysis and discussion to address the initial concerns. We thank the reviewers for the constructive criticisms as we feel that the manuscript has been improved significantly.

Reviewers' comments:

Reviewer #1 (Remarks to the Author):

In the manuscript “SOX17 regulates uterine epithelial-stromal crosstalk acting via a distal enhancer upstream of *Ihh*”, authors have used genetic mouse models and sequencing analysis to identify the function of transcription factor Sox17 in uterine receptivity. Authors claim that Sox17 regulates uterine epithelial-stromal crosstalk through a distal enhancer upstream of *Ihh*.

Major concerns:

Comment 1: This reviewer's biggest concern is that authors do not make a very clear distinction, if the function of Sox17 is essential in luminal epithelium or glandular epithelium or both. Most of the immunohistochemical data focuses on the luminal epithelium as uterine receptivity is a function of luminal epithelium. While this reviewer agrees that Sox17 is essential for endometrial receptivity, this reviewer feels that the role of glandular epithelium has been overlooked. Given the author's data there are three pieces of evidence that point out that this is at least as much a gland phenotype as luminal epithelium receptivity phenotype:

- a) The Sox17 array overlapped with the Foxa2 array – evidence that this is a gland phenotype not just a general epithelium phenotype.
- b) The Sox17 binding peak was also bound by FOXA2 and since FOXA2 is only in the glandular epithelium this would suggest that the gene set essential for receptivity and the enhancer region identified by the authors is important in glandular epithelium as opposed to the whole epithelium.
- c) When Sox17 binding enhancer is deleted upstream of *Ihh*, FoxA2 expression is intact in the glands and there is a partial fertility phenotype.

In order to address the separate function of Sox17 in glandular epithelium vs luminal epithelium and to also compare further this phenotype with the FoxA2 deletion phenotype, authors should inject LIF at the appropriate time and determine if LIF supplementation alone will rescue part of the phenotype. If in case there is complete rescue (like in the case of *LtfCre;FoxA2* deletion) then the phenotype is completely a gland phenotype. If there is a partial rescue that will support Sox17 having a role both in glandular epithelium and luminal epithelium.

Response:

We performed the LIF rescue as suggested by both reviewers to determine the potential contribution by the SOX17's glandular function on uterine receptivity. Five *Sox17^{f/f}* and five *Sox17^{ed/ed}* female mice each received two intraperitoneal injections of either vehicle or LIF (10µg), one at 1000 hour and one 1800 hour of GD 3.5. Uteri were collected on GD 5.5. *Foxa2^{d/d}*

female (n=3) that received LIF injection served as positive control. LIF fails to rescue implantation in the mice with uterine *Sox17* deletion although it was capable of rescuing the implantation phenotype in the control *Foxa2*^{d/d} mice (Supplementary Figure 1i as shown below).

“Supplementary Fig. 1 i, Embryo implantation sites (IS, indicated by arrow) were observed on GD 5.5 in vehicle-treated *Sox17*^{ff} mice and in LIF-replaced *Pgr*^{Cre/+}*Foxa2*^{ff} (*Foxa2*^{d/d}) mice but neither in vehicle-treated nor LIF-replaced *Sox17*^{ed/ed} mice (n=5 mice per group; for *Foxa2*^{d/d} mice, n=3).”

This result shows that rescuing glands functions by LIF supplementation is not sufficient to reverse the implantation defect, which supports that *Sox17* regulates other pathways, *Ihh* for one, that are critical for implantation. As the reviewer pointed out, uterine glands in the mice with *Sox17* ablation are altered; and the major impact of *Sox17* deletion on the uterine glands were seen by their absence FOXA2 and LIF expression in the *Sox17*^{ed/ed} mice and by the significant overlap with transcriptomic data on mice with *Foxa2* deletion in the uterus. While the present models are unable to isolate the glandular function of *Sox17*, our findings may inspire the scientific community to conduct future investigations, perhaps, by gland-specific *Sox17* ablation models.

Based on the new data, this manuscript has been modified as described below. (see Lines 140-151, 488-492, 756-758, Supplementary Fig. 1i)

Lines 140-151:

“Previous work has demonstrated that loss of uterine glands results in implantation defects and that the implantation defect can be rescued by administration of recombinant LIF, rLIF, to pregnant mice. In order to determine if the implantation defect in mice with loss of Sox17 was solely due to a glandular defect and loss of LIF, we attempted to rescue the implantation defect by administration of rLIF. In this experiment, mice with deletion of Foxa2 using the Pgr^{Cre} mouse served as a control as this mouse has been shown to have an implantation defect which could be rescued by rLIF³⁴. Intraperitoneal injections of mouse rLIF at GD3.5 did not rescue embryo implantation in Sox17^{ed/ed} mice as compared to the positive controls (Sox17^{ff} mice receiving vehicle; and Foxa2^{d/d} mice receiving mouse rLIF) and the negative control (Sox17^{ed/ed} mice receiving vehicle) (Supplementary Fig. 1i). This demonstrates that although LIF expression is dependent on SOX17, other pathways regulated by SOX17 are critical for implantation in the mouse uterus.”

Lines 488-492:

“LIF rescue assay

The attempt to rescue the implantation defect in the Sox17^{ed/ed} mice was using the protocol described by Kelleher and coworkers³³. Briefly, female mice received two intraperitoneal (i.p.) injections (one at 1000h and one at 1800h) of vehicle or 10µg recombinant mouse LIF on GD 3.5, and embryo implantation sites were evaluated on GD 5.5.”

Lines 756-758:

“Supplementary Fig. 1 i, Embryo implantation sites (IS, indicated by arrow) were observed on GD 5.5 in vehicle-treated Sox17^{ff} mice and in LIF-replaced Pgr^{Cre/+} Foxa2^{ff} (Foxa2^{d/d}) mice but neither in vehicle-treated nor LIF-replaced Sox17^{ed/ed} mice (n=5 mice per group; for Foxa2^{d/d} mice, n=3).”

Comment 2: Since Sox17 is a regulator of PGR action and Sox17 is also target of Pgr – how does using PGRcre by itself affect Sox17 expression? Is the Sox17 d/d phenotype exaggerated by loss of one copy of PGR? The authors should comment on this in the manuscript.

Response:

We conducted immunohistochemistry to examine the expression of SOX17 in wild type and Pgr^{Cre/+} mice. As shown in *Supplementary Fig. 1 d* (as seen below), no differences in SOX17 expression were detected in uteri between Sox17^{ff} and Pgr^{Cre/+} mice at GD3.5. Also, compound heterozygous mutations of Pgr^{Cre} and Sox17^{ff/+} did not produce any phenotypes that were observed in the Pgr^{Cre/+} Sox17^{ff}; and the Pgr^{Cre/+} Sox17^{ff/+} females remain fertile (see Response to Comment 1 by Reviewer 2.). In addition, both Pgr^{Cre/+} and Ltf^{Cre/+} driven ablations of Sox17 resulted in the same phenotype, i.e., failure of implantation and decidualization. These findings collectively suggest that loss of a single copy of Pgr does not contribute to the Sox17^{d/d} phenotype.

The text has been modified as below:

Lines 90-92:

“Also, no differences in SOX17 expression were detected in uteri between Sox17^{ff} and Pgr^{Cre/+} mice at GD3.5 (Supplementary Fig. 1d).”

Lines 747-749:

“Supplementary Fig. 1 d, Immunohistochemical staining of SOX17 in uteri between Sox17^{ff} and Pgr^{Cre/+} mice (n=3 mice per group) at GD3.5. Scale bars, 100µm.”

Comment 3: The authors choose to analyze expression of Sox17 in a pseudopregnant uterus. The rationale is unclear to this reviewer as to why a normal pregnant uterus was not evaluated.

Response:

We chose this approach because a pseudopregnant preimplantation uterus is physiologically similar to a normal pregnant uterus before GD5.5 and murine embryo did not implant until late GD4.5 (GD5). Since the pseudopregnant uterus did not contain embryo, it would provide clear results for concurrent molecular assays. However, in compliance with the reviewer, we have performed the SOX17 staining in the uteri during normal pregnancy as suggested. The results of this analysis showed the same pattern of SOX17 expression as in the pseudopregnant uteri. We have replaced the SOX17 expression pattern during the pseudopregnancy with the one from a normal pregnant uterus. (see Lines 74-75, 740-742; and Supplementary Fig. 1a). The manuscript has been modified as below:

Line 74-75:

“Gestation Day (GD) 0.5 to 4.5, were assayed by immunohistochemistry in the pregnant wildtype female mice (Supplementary Fig. 1a).”

Lines 740-742:

“a, Temporal and spatial expression of SOX17 in cross-sections of the adult mouse uterus throughout the preimplantation period, Gestation Day (GD) 0.5 to 4.5 (n=3 mice per group).”

Comment 4: *In the Sox17 deletion mutant – authors suggest that loss of Sox17 leads to reduction of PGR in the stromal cells – is this a reduction or since PGR is induced in the stromal cells at this time in pregnancy, there is lack of induction of PGR in the stroma?*

Response:

The reviewer is correct. It is not the reduction in PGR in the stroma but lack of induction. We have made that correction as stated below.

Line 178-179:

“Progesterone receptor (PGR) levels are reduced in the epithelial cells, and are lower in the stromal cells due to lack of induction during the preimplantation period.”

Comment 5: Is Arid1a expression down regulated in the Sox17 dataset?

Response:

Arid1a mRNA levels did not change in the *Sox17* microarray dataset, but reduction of protein was observed in *Sox17* deleted uteri by IHC. The down regulation of ARID1A could be due to the posttranslational regulation. The text has been modified as below.

Lines 227-230:

“Although Arid1a mRNA level did not change in the Sox17 KO microarray, immunohistochemical analyses showed the reduction of ARID1A in the Sox17^{ed/ed} uteri as compared with Sox17^{fl/fl} uteri at GD 3.5 (Fig. 4d, top panel), which could be due to the posttranslational regulation.”

Comment 6: Since the microarray matched up to both FoxA2 and Arid1a, this phenotype is

likely a common gene expression signature in the absence of receptivity or pregnancy. This should be emphasized more in the manuscript and perhaps in the title.

Response:

Among *Sox17*, *Foxa2* and *Arid1a* microarrays (MAs), 161 common genes were identified, which was 33.4% of 482 overlapping genes between *Sox17* and *Foxa2* MAs, and 46.8% of 344 overlapping genes between *Sox17* and *Arid1a*. Thus, gene expression signatures of *Foxa2* and *Arid1a* may involve independent pathways. Ablation of *Sox17* impacts *Foxa2* expression in the glands (Fig. 1c) and *Arid1a* expression in the stroma (Fig. 4d). Therefore, ablation of *Sox17* in the uterus has an impact on both compartments. While our findings demonstrate the importance of these pathways in the *Sox17* regulatory network, we are not sure whether including them into title is appropriate given that the common 161 genes account for a relatively small subset of genes regulated by the *Sox17* network. We therefore have respectfully chose not to alter the title.

Comment 7) This reviewer is confused on why the CHIP was performed with Ovx and P4. Is there PGR in the uterine epithelium in mice that have been ovariectomized and kept for two weeks? If not, then how will the P4 activate the PGR specific gene set?

Response:

Epithelial PGR levels are maintained in ovariectomized female mice two weeks post surgery. In this system, PGR genomic occupancy can be induced within 1h and expression of majority of PGR-regulated genes can be altered within 6 hours after P4 injection. This is the conditions in which we identified *Sox17* as a potential regulator of uterine gene expression and the same conditions we conducted ChIP-ChIP analysis for SOX17 in the uterus. (Rubel et al., Mol. Endocrinol 2012. PMID: 22638070)

Comment 8) In order to confirm loss of *Ihh* specifically in the uterus, the authors need to show *Ihh* protein expression loss by IHC or IF in the *Ihh* enhancer deleted uteri.

Response:

IHH expression was evaluated by IHC in uteri and kidneys between *Ihh19^{d/d}* and *Ihh^{+/+}* mice at GD 2.5. As results, uterine IHH were significantly reduced in *Ihh19^{d/d}* mice while remain the same level in the kidney as *Ihh^{+/+}* mice. (see Lines 306-309, 843-847; Figure below and Supplementary Fig. 4a)

Therefore, ablation of the enhancer at 19kb reduces but not ablates *Ihh* expression in a uteri specific fashion. This explains why ablation of the 19kb enhancer does not completely phenocopy the *Pgr^{Cre/+} Ihh^{ff}* infertility phenotype. The text has been modified as below.

Lines 306-309

*“Immunohistochemical staining further confirmed the down-regulation of IHH in the *Ihh19*^{d/d} uteri at GD 2.5 (Supplementary Fig. 4a). However, the loss of IHH expression was not complete as cells positive for IHH could be identified in the *Ihh19*^{d/d} uteri at GD 2.5 (Supplementary Fig. 4a).”*

Lines 843-847:

*“Supplementary Fig. 4 | (related to Fig. 7) a. Immunohistochemical staining of IHH in GD 2.5 uteri and kidney from *Ihh*^{+/+} and *Ihh19*^{d/d} mice (n=3 mice per group). Normal goat IgG served as a negative control. Scale bars, 100μm.”*

Comment 9) The authors point to a role of Sox17 in epithelial – stromal communication – but the data is very weak in support of this claim and this is concerning as this is also a part of the title of the manuscript.

Response:

The premise of SOX17 regulation epithelial - stroma communication was based on our observation that SOX17 regulates *Ihh*. *Ihh* has been shown to be produced by the epithelium and impact gene expression and function in the stroma. We clearly show alterations in stroma gene pathways. This impact on the stroma then governs stroma to epithelial communication. A schematic figure summarizing the epithelial-stromal interaction via SOX17 and other implantation related molecules has been added to aid conceptual delivery of our findings (Lines 839-840; Fig. 8). SOX17 induces IHH in LE and GE, which acts on stroma via PTCH1/2, further increases COUPTFII and PGR in the stroma. Increases in stromal HAND2 inhibits production of FGFs sent to uterine epithelia. These paracrine signals mediate the crosstalk between epithelium and stroma (PMID: 26804062, 25023679) The text has been modified as below:

Lines 839-840:

“Fig. 8 | SOX17 regulates IHH signaling pathway to govern uterine epithelial-stromal interactions during the window of receptivity.”

Comment 10) Why is pESR1 staining the luminal epithelial cell membrane in control in Fig. 2a?

Response:

We included the IgG control slides in Fig. 2a to show that the membrane staining is due to non-specific staining of the pESR1 antibody (see Figure below). The manuscript has been modified as below:

Lines 762-763:

“IgG staining as the negative control.”

Comment 11) Fig 2a, sections, for Sox17 deletion used for ESR1 and pESR1 staining look very different from PGR staining? Is there a variable phenotype where luminal epithelium is sometimes pseudostratified and sometimes single layer columnar?

Response:

Figure 2a has been revised to display PGR, ESR1 and pESR1 immunoreactivities on adjacent sections for clarification on the structural consistency issue. (see Figure below and Fig. 2a in the manuscript)

Lines 761-763:

“a, Immunohistochemical staining of ESR1, phosphorylated ESR1 and PGR in GD 3.5 uteri from Sox17^{fl/fl} and Sox17^{ed/ed} mice (n=3 mice per group). IgG staining as the negative control.”

Reviewer #2 (Remarks to the Author):

An earlier publication using heterogeneous Sox17 KO mice revealed that critical role of Sox17 in embryo implantation (PMID 27053385). Another previous study revealed crucial role of Sox17 in uterine gland formation and fertility (PMID 27102016). It also demonstrated that Pgr-Cre/Sox17-loxP (uterus-specific Sox17 KO, Sox17d/d) mice and Sprr2f-Cre/Sox17-loxP (uterine epithelium-specific Sox17 KO) mice are infertile and FoxA2 is a target of uterine Sox17, indicating epithelial Sox17 is essential for embryo implantation. Based on these studies, the current study by Wang X et al. confirmed that implantation is impaired in mice with epithelium-specific deletion of Sox17 (Ltf-iCre/Sox17-loxP, Sox17ed/ed). Importantly, this study newly found that uterine Sox17 regulates the implantation-related factors and pathways such as FoxA2-LIF pathway and Ihh signaling pathway. Notably, this study showed that mice with deletion of Sox17-binding site in Ihh locus are subfertile with reduced uterine induction of its downstream genes, indicating Sox17 directly regulates Ihh in the uterus. Although the current study includes some novel findings, it is unclear what is conceptual advance provided by this study. All the targets which authors provided are well-known factors and pathways in the process of implantation. Which pathway driven by Sox17 is most critical for implantation? Can LIF supplementation recover implantation failure in Sox17ed/ed mice? I think contribution of stromal Sox17 is not negligible, so the authors should examine the role of stromal Sox17 in vivo.

Response:

We respect the comments of Reviewer 2 and would like to address these issues in four parts.

1. With respect to the conceptual advancement, this manuscript delineates the *Sox17* dependent signaling pathways in mouse models by transcriptomic, cistromic and clinical data and further show the human relevance of SOX17 in normal and endometriotic endometrium. We demonstrate that altered *Sox17* expression not only impacts the expression of uterine genes but also impacts uterine epithelial differentiation. Most importantly, using the CRISPR-Cas9 mediated *in vivo* genome editing technology, we have functionally characterized a uterine specific enhancer that may be translated to other uterine epithelial genes and shed light on the regulation of uterine epithelial gene expression.

2. The ability of LIF to rescue the phenotype has been addressed in the response to Comment 1 by Reviewer 1. In brief, LIF fails to rescue implantation defects in the *Sox17^{ed/ed}* females although it was capable of rescuing the implantation phenotype in the control *Foxa2^{d/d}* mice (Supplementary Figure 1i)

3. Hirate and colleagues (PMID: 27053385) indicate that the *Sox17KO* heterozygotes were subfertile due to implantation defects. However, this paper did not demonstrate this phenotype was of uterine origin. In our study, *Pgr^{Cre}Sox17^{f/+}* produced an average litter size of 5.3 ± 0.79 pups/litter while *Sox17^{f/+}* females gave litter size of 5.9 ± 0.89 pups/litter. This is distinctly different from the previous report. One potential explanation on the phenotype difference could be the GFP allele in the *Sox17* locus. It has been shown that high levels of GFP by itself are able to confer an adverse phenotype (PMID: 10802676). In the case of *Sox17* GFP knock-in mice, it is not clear whether loss of one copy of *Sox17* and the GFP expression had a synthetic impact on the manifestation of the reported phenotype. We have modified our text as below to include the aforementioned new data:

Lines 361-363:

*“This was not observed in the *Sox17^{d/+}* or *Sox17^{ed/+}* in this study, in which *Sox17^{d/+}* (*Pgr^{Cre/+}Sox17^{f/+}*) produced an average litter size of 5.3 ± 0.79 pups/litter ($n=5$ dam) while *Sox17^{f/+}* females gave litters of 5.9 ± 0.89 pups/litter ($n=4$ dam), suggesting...”*

4. Guimarães-Young and colleagues (PMID: 27102016) describe the impact of the *Sox17* ablation in the uterus in which uterine epithelium defects impair fertility by inhibiting gland formation and/or FOXA2 expression. Their paper also suggests a role for stromal SOX17 for the uterine receptivity. While our findings agree with the overall infertile phenotype of their studies, we demonstrated the underlying mechanisms, as stated previously, that are distinct from what Guimarães-Young and colleagues suggested. Their conclusion that the remnants of the glands in the *Sprr2f-Cre/Sox17-loxP* are due to lack of *Sox17* ablation in stromal cells is interesting. However, in our studies, we found that the stromal SOX17 immunoreactivity is colocalized with endothelial cells (Supplementary Fig. 1c). Moreover, we further show that the *Pgr^{Cre}* mouse does not ablate gene in the vasculature, SOX17 proteins can be seen in the stromal vasculature of *Sox17d/d* mice (Supplementary Fig. 1b,c), and *Sox17d/d* females exhibited defective uterine phenotypes. Our data supports the fact that ablation of *Sox17* in the neonatal uterus results in no uterine glands. At puberty, once the glands are developed ablation of *Sox17* does not alter gland

development but the expression of FoxA2 in the glands. A similar phenotype has been reported by Kelleher et al (PMID: 28049832) when FoxA2 was ablated in the neonatal versus adult mouse uteri.

Specific comments are as follows:

Comment 1: In Line 328, the authors described that epithelial deletion of Sox17 using Sprr2fCre mice are fertile. According to the reference (PMID: 27102016), their wrong description might be due to their misunderstanding. Mice with Sprr2fCre-induce epithelial deletion were infertile, but did not impair gland formation. Reproductive phenotypes in this reference are similar to those shown by the current study.

Response:

We have changed the description in the manuscript accordingly.

Lines 61-65

“Similarly, ablating Sox17 in uterine epithelia also results in infertility²⁸. Here, we provide mechanistic insights on SOX17 regulation of fertility where epithelial SOX17, via regulation of Indian hedgehog Ihh expression, governs uterine epithelial cell proliferation, the ability of the epithelium to support embryo implantation, and the subsequent decidualization of endometrial stroma cells.”

Lines 363-374

“Sox17 using the Sprr2f^{Cre} mice are infertile with normal adenogenesis²⁸. Our findings further show that ablating Sox17 in the uterine epithelium by the Ltf^{Cre} results in complete abolition of embryo implantation, which not only agrees with the previous report but also provided a more detailed functional timeline of epithelial Sox17 on fertility. The Sprr2f^{Cre} mediated Sox17 knockout model also show the presence of glands in the uterus, which differs from that of the Pgr^{Cre} mediated Sox17 ablation model²⁸. It was speculated that the persistence of glands was due to an involvement of SOX17-expressing stromal cells. Our results identified the SOX17-expressing cells in the stromal compartment as endothelial cells. Moreover, deleting Foxa2 in uterine epithelium after puberty also allows establishment of glands despite of sterility³⁴. Therefore, the presence of glands in both Sprr2f^{Cre} and Ltf^{Cre} models are likely due to the timing of Sox17 ablation. In both models, gene recombination is delayed till after puberty^{30,34,49}.”

Comment 2: In Figure 1c, Ki67 staining in Sox17d/d mice look almost normal, while epithelium-specific Sox17 deletion promotes epithelial proliferation in Sox17ed/ed mice. How does stromal and epithelial Sox17 control cell proliferation and differentiation? Does stromal deletion of Sox17 normalize epithelial differentiation?

Response:

Quantification of Ki67 staining was performed by H-score analysis, which showed that Ki67 was highly expressed in the LE of both Sox17^{ed/ed} and Sox17^{d/d} uteri as compared to Sox17^{fl/fl} mice at GD 3.5. (see Lines 119-126, 755-760; and supplementary Fig. 1h). The increase in proliferation

in the $Sox17^{d/d}$ is significantly greater than the control. Reviewer 2 is correct in noticing that Ki67 staining in the $Sox17^{ed/ed}$ is significantly greater than that of $Sox17^{d/d}$. This change in proliferation is unexpected and may be due to the neonatal versus adult ablation of $Sox17$ or the changes in the uterine epithelium due to the focal stratification phenotype. We have not attempted stroma ablation for $Sox17$ because it is our vast experience that there is no uterine stroma Cre model that is specific for this compartment of the uterus.

The text has been modified as below

Lines 116-123:

“Analysis of uterine epithelial cell proliferation, as measured by Ki67 staining, shows that at GD 3.5 $Sox17^{fl/fl}$ mice exhibited minimal Ki67 staining (H-score; 2.86 ± 2.00) while both $Sox17^{ed/ed}$ (H-score; 217.30 ± 9.68) and $Sox17^{d/d}$ (H-score; 179.10 ± 6.03) uterine epithelium stained positive ($P < 0.05$) for Ki67 and thus did not exhibit the inhibition of proliferation during the preimplantation period (Fig. 1c, top panel and Supplementary Fig. 1h). Interestingly, Ki67 staining in the $Sox17^{ed/ed}$ is greater ($P < 0.05$) than that of $Sox17^{d/d}$. This change in proliferation is unexpected and may be due to the neonatal versus adult ablation of $Sox17$ or the changes in the uterine epithelium due to the focal stratification phenotype.”

Lines 753-758:

“h, H-score quantification of Ki67 in endometrial section of $Sox17^{fl/fl}$, $Sox17^{ed/ed}$ and $Sox17^{d/d}$ mice ($n=3$) at GD 3.5. Different superscript letters denote significant ($P < 0.05$) differences. Data are presented as means and SEM.”

Comment 3. The authors previously showed importance of P4-Pgr-Ihh signaling in uterine receptivity. The present study showed that deletion of $Sox17$ reduces the expression of Pgr and

Ihh. Is the downregulation of Pgr a major cause of dysregulation of Pgr signaling in Sox17ed/ed mice? In Figure 2a, how is stromal expression of Pgr downregulated in Sox17ed/ed mice?

Response:

Based on the transcriptomic and cistromic datasets, SOX17 and PGR co-regulated the PGR signaling and IHH signaling. Lower levels of stromal PGR were due to the impaired IHH signal transduction from epithelia to stroma, subsequently affecting the stromal COUPTF-II that serves as the regulator of stromal PGR (see Fig. 8).

Fig. 8

The text has been modified as below:

Lines 342-346

“Ablation of Sox17 or the SOX17 binding region distal to the Ihh gene alters proper epithelial-stromal crosstalk in the uterus required for pregnancy^{3,40}, in which SOX17 and PGR co-regulated the PGR signaling and IHH signaling. The impaired IHH signal transduction from epithelial to stroma, subsequently affected the stromal PGR via its regulator, the stromal COUPTF-II (Fig. 8).”

Lines 839-840:

“Fig. 8 | SOX17 regulates IHH signaling pathway to govern uterine epithelial-stromal interactions during the window of receptivity.”

Comment 4: It is difficult to understand the epithelial-stromal interaction through Sox17 and other implantation-related molecules, because so many implantation-related molecules were examined with no distinction between stromal and epithelial specific genes in this paper. For

example, Supplemental figure 4 contains both epithelial and stromal specific genes. I struggled to understand the relationship among these key factors for implantation, especially in the latter half of manuscript. Please edit the text and make it understandable. Schematic figures might be useful.

Response:

We have added a Schematic figure summarizing the epithelial-stromal interaction via SOX17 and other implantation related molecules has been added as suggested (see Lines 842-843; Fig. 8).

Lines 839-840:

“Fig. 8 | SOX17 regulates IHH signaling pathway to govern uterine epithelial-stromal interactions during the window of receptivity.”

Comment 5: In Figure 5a, are the endometriosis samples endometriotic lesions, or eutopic endometria in the patient with endometriosis? Are the endometria of proliferative and secretory phase obtained from women with or without endometriosis? Background of human samples should be described in detail. how many human samples were investigated in each group? Quantification of immunostaining and statistical analysis should be recommended.

Response:

The endometriosis samples are eutopic endometria in the patient with endometriosis. Endometria of proliferative and secretory phase were obtained from women with and without endometriosis (n=7 per phase per control or endometriosis). H-score analyses were performed as suggested. (Lines 236-244, 425-433, 508-809, 791-794). The description has been modified accordingly.

Lines 236-244:

“To investigate SOX17 role in human endometrial function, we analyzed its expression pattern in endometrial tissue from women with and without endometriosis at proliferative and secretory phases (n=7 per phase per disease). SOX17 was highly expressed in the endometrial epithelia of women without endometriosis at the proliferative (H-score; 288.00 ± 8.25) and secretory phases (H-score 278.57 ± 9.01), However, its expression was significantly decreased in the endometrial epithelia of women with endometriosis at the proliferative (H-score; 119.29 ± 38.83) as well as secretory phases (H-score 48.57 ± 12.94) compared with controls (Fig. 5a,b). This demonstrates that the uterine epithelium expression of SOX17 is conserved between mouse and human and in the human endometrium SOX17 expression is impacted by endometriosis.”

Lines 425-433:

“Human endometrial samples

This study was carried out in accordance with federal regulations governing human subjects research. All procedures were approved by the following ethics committees: Institutional Review Board/Committee-A (IRB) of Greenville Health System under IRB file # Pro00000993 and Pro00013885 and the University of Chapel Hill at North Carolina IRB under file #: 05-1757.

Informed consent was obtained from all patients before their participation in this study. To examine SOX17 expression patterns of eutopic endometrium between women with and without endometriosis (control vs. endometriosis group), 28 samples in total were used from proliferative and secretory phases (n=7 per phase per group)."

Lines 508-509:

"A semiquantitative grading system (H-score) was used to compare the immunohistochemical staining intensities as previously described⁶⁰."

Lines 791-794:

*"Immunohistochemical staining (a) and H-score quantification (b) of SOX17 in human endometrial section from women with and without endometriosis at proliferative and secretory phase of the menstrual cycle, as well as endometriosis patients. Scale bars, 50µm. *P<0.001. Data are presented as means and SEM."*

Comment 6. The figure and the legend of Figure 6c needs to be revised to make it understandable.

Response:

The figure and the legend of Fig. 6c has been revised as suggested. (see Figure below and Fig. 6c; and Lines 807-812)

Lines 807-812:

"c, Genome browser tracks of SOX17, FOXA2, PGR and GATA2 binding at putative enhancers for Ihh gene in P₄-treated or GD 3.5 uteri. UCSC Genome Browser views showing the mapped read coverage of SOX17, FOXA2, PGR and GATA2 ChIP-seq data. The Ihh gene locus and the regions targeted by gRNAs for CRISPR/Cas9 deletion in vivo are highlighted in blue and yellow, respectively. Ihh19, putative enhancer 19kb upstream of Ihh gene locus."

Comment 7: In Ihh19d/d mice, ovarian function including ovulation, fertilization and hormone production should be assessed.

Response:

As suggested, the number of embryo and the level of serum P4 at GD 2.5 were assessed, which showed no difference between *Ihh19^{d/d}* and *Ihh^{+/+}* mice. (see Lines 318-322, 845-847; Supplementary Fig. 4b and c)

Lines 318-324:

*“The decrease in implantation sites was not due to an impact on ovulation or embryo development as there was no difference in the number of embryo flushed from the uterus and oviduct at GD 2.5 between *Ihh19^{d/d}* and control mice at GD 2.5 (Supplementary Fig. 4b). There was no difference in the serum P4 level between these two groups, indicating a normal endocrine regulation of hormone secretion (Supplementary Fig. 4c).”*

Lines 845-847:

*“...b. The number of flushed embryo from *Ihh^{+/+}* and *Ihh19^{d/d}* mice at GD2.5 (n=15 mice per group). c. P4 level in serum from *Ihh^{+/+}* and *Ihh19^{d/d}* mice at GD2.5 (n=5 mice per group).”*

REVIEWERS' COMMENTS:

Reviewer #1 (Remarks to the Author):

The manuscript has been significantly improved and the new experiments add further clarity to the understanding of the role of Sox17 in uterine receptivity.

-Line 150 should be modified to read - although LIF expression is dependent on SOX17, other pathways regulated by SOX17 "in the glandular epithelium" are critical for implantation in the mouse uterus.

Reviewer #2 (Remarks to the Author):

The authors performed some experiments and revised the manuscript appropriately according to the comments of this reviewer.

Response to Reviewers for manuscript NCOMMS-18-06951A

REVIEWERS' COMMENTS:

Reviewer #1 (Remarks to the Author):

The manuscript has been significantly improved and the new experiments add further clarity to the understanding of the role of Sox17 in uterine receptivity.

-Line 150 should be modified to read - although LIF expression is dependent on SOX17, other pathways regulated by SOX17 "in the glandular epithelium" are critical for implantation in the mouse uterus.

Response

Thank you for the comment. The sentence has been modified as suggested by the reviewer as shown in line 155-157: "This demonstrates that although LIF expression is dependent on SOX17, other pathways regulated by SOX17 in the GE are critical for implantation in the mouse uterus".

Reviewer #2 (Remarks to the Author):

The authors performed some experiments and revised the manuscript appropriately according to the comments of this reviewer.

Response

Thank you.